# Sub-ångström resolution ptychography in a scanning electron microscope at 20 keV

Arthur M. Blackburn [1,2] ✉, Cristina Cordoba [1,2], Matthew R. Fitzpatrick [1,2] & Robert A. McLeod[3,4]

Achieving sub-ångström (<1 Å) resolution in electron microscopy typically requires a high-energy (>30 keV) beam and a transmission electron microscope (TEM) fitted with an aberration corrector and a high efficiency, high pixel-count camera. The cost, space- and personnel-requirements of such TEMs is prohibitive for many laboratories. Low-energy (≤30 keV) scanning electron microscopes (SEMs), in comparison, offer a simpler, more compact, and cost-effective electron microscopy platform. Lower beam energies also have the potential to provide greater information from thin, light-element samples. Here, we demonstrate a sub-ångström resolution of 0.67 Å using a 20 keV SEM operated in transmission mode, enabled by ptychographic reconstruction. This resolution corresponds to a resolution-to-wavelength ratio of 7.8, surpassing previous results from electron ptychography and conventional imaging approaches. The gain in resolution is achieved through a multi-slice ptychographic algorithm incorporating diffraction distortion correction, coupled with a SEM fitted with a cold field emission source, immersion lens, simple projector lens, and a hybrid direct detector optimized for low-energy electrons. This approach holds immediate promise for high-resolution imaging of 2D materials and, with further development, may enable structural studies of small (<100 kDa) proteins.

The information collected from specimen observations in an electron microscope increases with the spatial resolution of the microscope and the efficiency of its detectors. To achieve sub-Ångström (<$10^{-10}$ m) resolution for atomic resolution structural determination, current electron microscopes must generally use relatively high-energy electron beams (>30 keV), even when aided by recently developed aberration correctors[1]. Furthermore, when the observed sample is easily damaged under electron beam irradiation, as is the case with biological materials, the scattered electrons must be recorded with the greatest possible detection efficiency[2]. Consequently, much effort and work has been made in recent years to improve the detection efficiency of pixelated electron detectors for conventional transmission electron microscope (TEM) imaging modes and also for high-dynamic range,

high-speed scanning diffraction studies[3–5]. These detector advances, combined with recent advances in aberration correctors for TEM and cryogenic sample preparation methods, have been central in making TEM an invaluable and essential tool in modern microbiology, accelerating structural investigations related to pathogen biology, host-pathogen interaction and drug discovery, for example[2,6–8].

However, an aberration corrected (AC-) TEM with an advanced high-pixel count direct electron detector has a financial cost, space, and personnel requirement that is prohibitive for many laboratories[2]. In comparison, low energy (≤30 keV), non-aberration correction scanning electron microscopes (SEM) are smaller, have significantly lower cost and are more economical to run. Consequently, low-energy SEMs have become a ubiquitous and essential tool for micro- and

[1]Department of Physics and Astronomy, University of Victoria, Victoria, BC, Canada. [2]Centre for Advanced Materials and Related Technologies, University of Victoria, Victoria, BC, Canada. [3]Hitachi High-Tech Canada Inc., Toronto, ON, Canada. [4]Department of Materials Science & Engineering, University of Toronto, Toronto, ON, Canada. ✉e-mail: ablackbu@uvic.ca

nano-sciences, engineering, and technology. However, the resolution of a 30 keV non-aberration corrected SEM/STEM is generally at most around 4 Å for secondary electron imaging[9], and for STEM mode brightfield imaging, Si lattice fringes of 0.157 nm have also been observed[10]. While some aberration correctors and monochromators for ≤ 30 keV TEM[1,11] and SEM[12,13] have been developed in recent years to approach sub-Å resolution, they still retain the problems of cost and complexity. A relatively new approach to improving the resolution in TEM, which does not require an aberration corrector or a high-pixel count electron detector, is ptychography[14].

Ptychography operates on 2D diffraction data produced from many small overlapping sample illumination areas. It creates a quantitative model of the amplitude and phase shift accrued by the electron wave on passing through the sample using a computation that aims to provide the best match between modelled and collected diffraction data. The images it produces reveal information most similar to that produced from BF-STEM or TEM, though with the addition of quantitative phase-shift information. It is not like secondary electron (SE) imaging in the SEM: it does not reveal surface topography or give SE yield contrast. Ptychography requires the illumination to have significant coherence, and it falls within a wider group of coherent diffractive imaging (CDI) algorithms that see widespread use in coherent x-ray imaging[14]. Recent developments in high-speed hybrid-type pixelated direct electron detectors and advances in computational algorithms have enabled it to become a practical electron microscopy technique that can provide enhanced resolution and quantitative phase information[15]. For example, ptychography has been experimentally applied to biological samples in vitreous ice within a cryo-TEM[16] and to large-area imaging of samples at low magnification[17]. It has been also theoretically studied for applications in cryo-electron microscopy at low doses, potentially for sub 100 kDa macromolecular complexes[18]. Also, it was recently applied to electron microscopy of few-layer transition metal dichalcogenides (TMDCs, e.g., $MoS_2$, $MoSe_2$, $WS_2$) and thicker samples of $PrScO_3$[19–21] to give sub-Å information with an 80 keV[21] and 300 keV[19] beam energy, where it provided a 3 – 4x improvement over the conventional lateral resolution of the electron microscopes used in the experiments. The depth resolution using multi-slice ptychography is also significantly improved over conventional STEM imaging, going well below the 5 nm range[19,22]. This depth resolution capability has been advanced[23] and used to reveal depth-dependent crystal structure orientations around dislocation cores[22] and inhomogeneity in zeolites[24], for example.

However, the microscopes employed in these earlier sub-Ångström demonstrations of ptychography were already capable of sub-Å resolution, through using a high electron beam energy (≥ 80 keV) and an aberration corrector. Ptychography at much lower energies in non-aberration corrected SEMs appears relatively unexplored except for some of the first practical demonstrations of electron ptychography, which were performed with a 30 keV beam in a SEM[25]. In this first work, the extended ptychographical iterative engine (ePIE) algorithm[26] gave phase reconstructions showing the ⟨111⟩ atomic plane spacing in gold particles, which are known to have a spacing of 2.36 Å. This offered greater than 4 times resolution improvement for the SEM used in the experiments[25]. However, this work used, by today's standards, a low-efficiency and slow camera, along with a strongly constrained electron optical arrangement where the sample was physically glued to the objective lens pole-piece. Reprocessing the data from these early experiments using an algorithm to account for the non-ideal camera response yielded some phase reconstructions which perhaps indicated the ⟨022⟩ planes (with 1.44 Å separation) of gold particles, though this was not confirmed using standard resolution assessment methods[27]. More recent work using a 30 keV electron beam, in a 3rd order aberration corrected STEM as opposed to a non-aberration corrected SEM, formed a ptychographic reconstruction from padded diffraction data, demonstrated a resolution of at least

1.2 Å at 30 keV[28], thus providing a factor of 2 improvement on the first SEM-based demonstrations[25].

Aside from the economic reasons for adopting ≤ 30 keV electron beams for sub-Å imaging, a recent study pointed out that the greater elastic scattering cross section ($\sigma_e$) achieved with a lower energy beam gives an improved image contrast, which on balance, can outweigh the effects of increased sample damage as the sample becomes thinner at lower energies[29]. The information coefficient[29], defined as $\zeta = T(\sigma_e/\sigma_i)$, where $T$ is the total transmission through the sample and $\sigma_i$ is the inelastic scattering cross section, is optimised by reducing the beam energy as the sample thickness is reduced. For a sample composed primarily of light elements, such as carbon-based biological samples, and a beam energy ≤ 30 keV, this corresponds to a sample thickness ≤ 15 nm. This thickness region is applicable for technological devices and smaller natural and biological structures. If a protein is imaged while covered in an ice layer having a thickness comparable to its diameter, as is often used in cryo-EM[30], this would limit the protein diameter to approximately 5 nm. This in turn corresponds to protein molecular masses below about 50 – 100 kDa[31]. The capability of cryo-electron ptychography to study such low mass proteins at low electron doses has been theoretically studied[18], and appears promising in the light of recent experimental cryo-electron ptychography demonstrations[16,32]. These low mass proteins (< 100 kDa) are particularly abundant in nature but are difficult to characterise, and thus many have as yet undetermined structures[33]. Consequently, there is a strong drive and desire to determine the detailed structure of these proteins[34].

In addition, 2D materials, such as graphene, hexagonal boron nitride, $MoS_2$ and other TMDCs, which hold great promise for future electronics, generally rely on having sub-10 nm thickness active layers[35]. Many materials, such as $MoS_2$, have a primary knock-on displacement cross-section that diminishes rapidly below primary beam energies of 80 keV, but owing to electronic excitations, there is still a significant cross-section for displacement below this energy that only starts to diminish below 30 keV[36]. Such devices still require more research along with a scaling of production and associated observation techniques to meet the required levels for manufacturing readiness[37]. Furthermore, SEM instruments, which typically operate at ≤ 30 keV, have much greater chamber space around the sample in comparison to TEMs, which facilitates applying external stimuli and performing in situ experiments. Thus, many materials, technologies, and in situ atomic scale investigations would also benefit from sub-Å imaging at ≤ 30 eV.

## Results
### Diffraction data distortion correction

When a reconstruction or map of information in the sample plane is formed from diffraction-plane data, it is possible for errors and distortions in the diffraction data to corrupt the sample-plane information in non-obvious ways[38,39]. In ptychography, this can lead to reconstructions appearing plausible upon first inspection, while subsequent detailed analysis reveals physical inconsistencies. Here, to achieve sub-Å imaging, we aimed for a ~ 25 pm reconstructed pixel size, which thus requires collecting electrons scattered by up to ~ 10 degrees (~ 175 mrad) when using a 20 keV beam (see Supplementary Information, Equation S.2)[15]. In line with our goal of developing a high-resolution technique that could be applied to lower-cost general SEM/STEM class instruments, our projector lens system contained only a single lens (see Supplementary Fig. 10) that was optimised towards size, cost and convenience, rather than for distortions – the primary one being pincushion – which we could correct digitally. A well-designed, higher-voltage dedicated STEM would likely have less distortion than was the case with our experimental instrument.

Pincushion distortion is a primary aberration of round magnetic lenses, having a form where the ratio of the measured to actual distance from the centre of the diffraction plane varies in proportion to

the square of the distance from the optical axis[40] (see "Methods", Eq. 1). This relationship is similar in form and related to spherical aberration which operates on the image plane[40,41]. However, unlike spherical aberration in the electron-probe forming optics, which remains approximately constant over typical high magnification fields of view in the SEM, pincushion distortion over the diffraction plane cannot be represented by a singular point spread function that is applied to the diffraction data. Thus, such distortions on the diffraction plane cannot be absorbed into the probe reconstruction. Measurement and subsequent correction of the diffraction pattern distortion is thus essential to give meaningful and accurate reconstructions at the greatest resolution.

In conventional TEM instruments, radial distortion can be measured from a diffraction pattern collected from illuminating a known polycrystalline calibration sample, such as gold nanoparticles, with parallel or very low convergence angle illumination at the objective-lens excitation used for the data collection. The resulting sharp diffraction peaks can then be fitted to an expected radial distribution. However, conventional SEM instruments, including the instrument used in this work, generally have a more restrictive illumination or probe forming system than CTEM instruments, which often makes it impossible to illuminate the sample with parallel or low convergence angle illumination at the objective-lens excitation required to gain the required camera length and probe size. Thus, a convergent or divergent electron beam probe must be used when attempting to map diffraction pattern distortion. In this case, pincushion distortion manifests in the diffraction data as high-angle scattered diffraction discs appearing oval in shape as opposed to circular (see the discs marked A1, A2, C1 and C2 in Supplementary Fig. 1(a, c), and linearly measured scattering angles differing from their true values. Attempts to determine the distortion directly from this data using the non-uniform, weakly illuminated, non-circular discs in the diffraction patterns require significant subjective judgement and is hence unreliable and biased.

To overcome this challenge, and in an advance to existing reconstruction techniques and other diffraction pattern calibration techniques, we took advantage of the coupling between the diffraction pattern distortion and the reconstruction. We found that despite the diffraction data being distorted, a ptychographic reconstruction could be made, and its Fourier transform (FT), which we found by using the Fast FT algorithm, yielded well-defined low-order diffraction peaks concentrated in rings, as expected for a polycrystalline sample. However, the reciprocal-space coordinates of the ring positions differed from those expected for the ideal, non-distorted situation. We found that an ideal modelled radial diffraction profile for the sample could be computationally distorted using a simple pincushion model to give good agreement with experimental ring reciprocal-space coordinates (see Methods and Supplementary Fig. 2c). Consequently, we determined distortion coefficients that gave the best fit to the FT of the experimental reconstruction. These coefficients were subsequently used to undistort the diffraction data and the associated floating pixel mask.

A reconstruction created using the undistorted diffraction data had a reduced Fourier error metric, and the FT of the reconstruction contained stronger and sharper peaks. Further refinement of the distortion coefficient was then made by again fitting a distorted model diffraction profile to this FT, and the diffraction data was undistorted once more using this refined coefficient. Producing yet another reconstruction using this undistorted diffraction data (see process in Supplementary Fig. 2a) finally yielded an agreement between the expected and reconstructed reciprocal spacings ($1/d$) to within 0.5% out to the {135} diffraction ring of gold at 1.45 Å$^{-1}$ (= 0.689 Å), as opposed to the initial approximate 15% agreement before correction (see "Methods").

## Gold on amorphous carbon reconstruction

An example ptychographic reconstruction created from 20 keV electron diffraction data collected from gold particles supported on an amorphous carbon (Au/aC) thin film is presented in Fig. 1. To produce a physically valid sample object model, we used a multi-slice ptychographic reconstruction algorithm operating upon distortion-corrected diffraction data accompanied by a circular aperture constraint on the FT of to produce a physical probe model (see "Methods"). A plane wave was numerically propagated through the multiple slices of the resulting reconstructed object model to give the phase and amplitude (Fig. 1b) of a reconstructed exit wave. Presenting a propagated exit wave, rather than a summed projected phase or product of amplitudes, gives images similar to those produced by bright-field CTEM. These exit-wave images produce a FT that has the expected sequence of diffraction ring intensities for randomly oriented particles, in contrast to summing the projected phase from all slices (as discussed in Supplementary Information, section 3, with comparisons presented in Supplementary Figs. 6 and 11). Looking at an enlarged view of two merged Au particles (Fig. 1b), we see features familiar to conventional AC-TEM observations of Au particles, such as $n$-fold twinning[42,43]. From

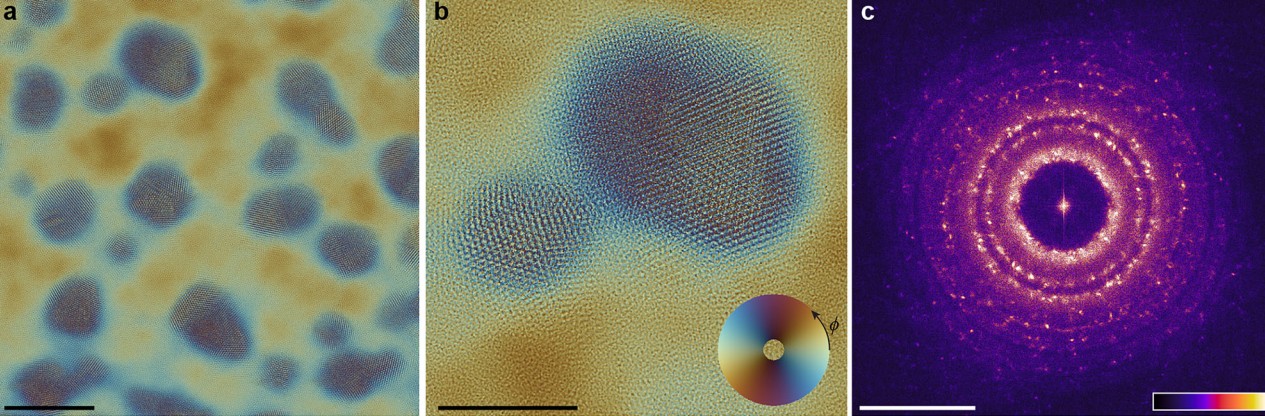

**Fig. 1 | Gold on amorphous carbon reconstruction results. a** Exit wave produced by simulating the propagation of an electron plane wave through the multi-slice model produced by a ptychographic reconstruction on experimental 20 keV electron diffraction data obtained from gold particles supported on amorphous carbon, and (**b**) enlarged region of the reconstructed exit-wave. The colour scale for (**a** and **b**) is in the lower right corner of (**b**), where the colour wheel's azimuthal angle, $\phi$, represents the phase (spanning $0 - 2\pi$) and the radial coordinate represents the amplitude of the wave, spanning an amplitude of 0.43 – 2.35. **c** Amplitude of the Fourier transform (FT) of the intensity of the reconstructed exit wave. In (**c**), the intensity is gamma adjusted by 0.9, the upper and lower 1% of pixels are saturated, and the colour scale, which linearly represents the amplitude$^{0.9}$ is inset at the lower right. Scalebars: (**a**) – 10 nm, (**b**) – 5 nm, (**c**) – 1 Å$^{-1}$.

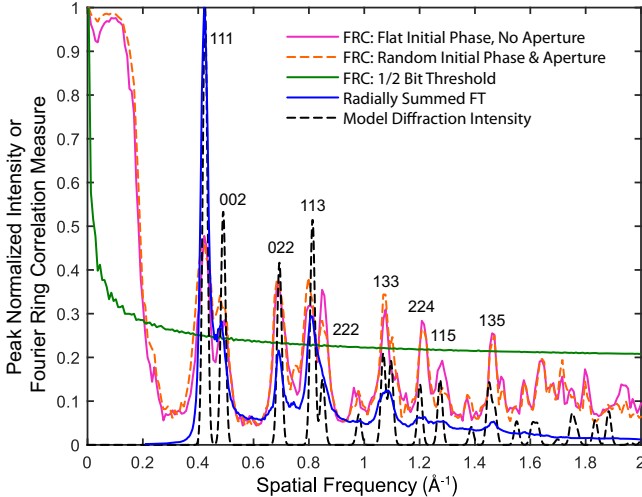

**Fig. 2 | Fourier ring correlation and reconstruction radial profiles from gold on amorphous carbon.** (Blue line) Radially summed profile of the FT of the gold on amorphous carbon reconstruction given in Fig. 1c; (dashed black line) model radial diffraction intensities expected for randomly oriented gold particles, labelled with dominant diffraction plane indices for each peak; (magenta line) Fourier ring correlation (FRC) measure obtained from comparison of two independently determined reconstructions, each using half of the data used to form the reconstruction in Fig. 1; (dashed orange line) FRC from a reconstruction using an aperture on the probe FT and an initial random phase object; and (green line) the half-bit resolution threshold. The maximal crossing of the FRC characteristic with the threshold gives the resolution as 0.67 Å.

the FT of the reconstruction (Fig. 1c), we see diffraction rings and spots arising from {135} planes in the particles, which have a spacing of 0.689 Å[44].

The resolution of the reconstructions produced from this sample was assessed using the Fourier ring correlation (FRC) metric[20,45,46]. This metric analyses two independent reconstructions or images to give a degree of correlation with respect to spatial frequency ("Methods", and Supplementary Information, section 5). The maximum spatial frequency at which the FRC is above a threshold level determines a measure of the image resolution[46]. Here we take the ½ bit threshold, which corresponds to a flat signal-to-noise ratio (SNR) of 0.41 in the Fourier domain[45], as has been used in earlier assessments of ptycho-graphic resolution[17,18,45,47] (see also "Methods"). Here, we see (Fig. 2) that the FRC characteristic corresponding to the Au {135} plane spacing at 1.45 Å$^{-1}$ ( = 0.689 Å) clearly correlates above the threshold, with and without a constraint on the probe FT and with constant and random initial object guesses (see "Methods"). Following the {135} plane correlation peak, the FRC finally intersects the ½ bit threshold at 1.484 Å$^{-1}$, giving a resolution measure of 0.67 Å.

In Fig. 2, we additionally plot the average radial profile of the FT of the reconstructed intensity and the diffraction intensity profile expected from randomly oriented gold particles, where we label the dominant Au diffraction peak indices. The non-crystalline structure of the amorphous carbon, where the structural information is broadly spread over a wide spatial frequency range, would require a greater electron dose to reliably determine its structure and show correlation between independent scans in comparison to crystalline gold which concentrates most of its scattered intensity into or near discrete peaks (or Bragg discs) that are easier to distinguish in the presence of inevitable dose-limited shot noise. Furthermore, the high electron beam intensity in a CFEG STEM system can readily induce structural rearrangement and instability in aC, through mechanisms such as electron beam sputtering and deposition[48], which further reduces the likelihood of correlations between reconstructions of amorphous carbon.

These factors make a stable reconstruction of the aC much less likely that the gold particles, which, in comparison, remain stable in the electron beam. Thus, the low FRC measured at spatial frequencies between those of the Au diffraction peaks here is not necessarily a deficiency in the algorithms to reconstruct at these spatial frequencies. Indeed, recent cryo-ptychographic work, when optimised towards conditions for low-spatial frequencies, has demonstrated good information transfer across the low spatial frequencies present in amorphous carbon[32].

Coherent electron scattering from the amorphous carbon support and any amorphous overlying contamination layer may play a role in helping explain the high FRC scores we see at resolutions beyond what would be expected from information in the bright field disc, while operating in this so-called darkfield mode of electron ptychography used here. At the low beam energies we used, dynamical (multiple) scattering of the electron beam is more likely than at the higher energies typically used in TEM. Thus, with an overlying aC-like layer, when the electron beam reaches buried crystalline regions, it no longer has the form of the electron beam in the vacuum: its angular form extends beyond that of the original brightfield disc of a direct, un-scattered beam. Similarly, an underlying aC-like support layer will multiply diffract beams scattered from the crystalline regions, potentially causing them to interfere. Thus, relative phase information between two otherwise separated crystalline diffracted beams could possibly be determined from the diffraction data using ptychography. Put another way, the support film and contamination layers diffract the illumination into a range of spatial frequencies which is much broader than the discrete peaks of the polycrystalline gold regions, and we posit this may help explain the repeatability we see in the phase of the reconstructions of sample crystalline regions, at spatial frequencies beyond that expected from simple consideration of the brightfield disc. Without this mechanism or other means to consistently determine the relative phases between otherwise separated diffraction discs, ambiguity in reconstructed phases may result.

While a further study involving multi-slice simulations would be required to completely confirm this proposed mechanism, here we simulated the limiting case of a single-slice reconstruction of a simple mixed-phase and amplitude object having a broad range of spatial frequencies and isolated resolution-testing double-dot like features (see Supplementary Fig. 8a, b). The simulation used a beam convergence angle, defocus, illumination overlap, and reconstructed pixel size which were similar or identical to the experiment (see Supplementary Information). The simulated reconstruction has a phase contrast transfer function (PCTF) characteristic (Supplementary Fig. 8e) with a steady near-unity value out to spatial frequencies ~1.5 Å$^{-1}$, and an SNR (Supplementary Fig. 8f), which diminishes from ~10$^4$ at low spatial frequencies to 10 at just over ~1.5 Å$^{-1}$. Noting the relationship between FRC and SNR[49,50], the ½ bit FRC threshold would correspond to an SNR of 0.41, which appears at ~2.2 Å$^{-1}$ in these reconstructions.

Our experimental reconstructions, however, use a multi-slice approach, with a number of slices and separation that is towards the outer bounds of yielding a useful approximate multi-slice reconstruction[51]. The finite experimental electron dose, possible beam positional error, and the relatively few slices in our reconstruction likely contribute to our experimentally estimated repeatable information limit being less than from the single-slice simulation.

Ptychography also gives us a model of the electron probe. This shows the illuminating probe beam to have a convergence semi-angle of 7.2 mrad and an approximate diameter of 4 nm on the sample (Supplementary Fig. 3). Furthermore, the primary aberrations of the electron probe forming system were extracted from the probe model (Supplementary Fig. 5, and Supplementary Information, section 2). This revealed a spherical aberration coefficient of 3.9 mm, which agrees approximately with the expected value for our objective

lens excitation and sample position, further validating the reconstruction, and shows the probe was defocussed by − 390 nm. If our electron probe was at focus, the diffraction-limited resolution as determined by the Rayleigh criterion and our probe convergence semi-angle would be 7.3 Å, which is 10.9 times larger than our measure of the ptychographic resolution. Defocused probe electron ptychography, where the dark-field diffraction data is also processed in the ptychographic reconstruction (as used here), does not require a high-resolution, finely focused electron probe, which is typically used in high-resolution analytical STEM and SEM. However, we note that defocusing the probe would also lower the resolution of x-ray spectra mapping if it is performed simultaneously with the ptychographic data acquisition.

When a focused probe is used, with a convergence semi-angle sufficient to provide overlap between all diffracted beams, the depth resolution is improved and has been shown to surpass that of conventional STEM[19,22], heading below ~ 5 nm. In our work, we have not directly determined the depth resolution, as our goal was to produce TEM-like 2D images of the sample, rather than full 3-dimensional models. Determination of our depth resolution would require further analysis and modelling, which will be part of future work.

## Gold on MoS₂ reconstruction

As an example 2D TMDC material, we observed a few-layer $MoS_2$ decorated with Au islands. This has previously been used as a model system to investigate Moiré engineering between 2D and 3D materials[52]. Here, the $MoS_2$ has a uniform crystalline structure with an essentially single orientation, excepting small local variations in tilt due to sample waviness. Consequently, with our use of a defocused electron probe, its diffraction patterns have a relatively high degree of similarity to each other and thus reduced diversity in comparison to the Au/aC sample. Thus, we anticipated this $MoS_2$ based sample to present a stronger test of our low-energy ptychography methodology, as other studies have noted the importance of data diversity in achieving reliable reconstructions[14].

To form a convergent and physical ptychographic reconstruction (Fig. 3a, b), as with the Au/aC reconstruction, we found it necessary to force pixels outside a certain radius in the FT of the electron probe model towards zero to gain a stable, convergent and physical probe and object reconstruction (see "Methods"). Regarding the probe reconstruction, we note that ptychography in its simplest and first developed forms assumes a single coherent illumination mode propagates elastically through the sample[14]. In practice, real illumination systems have limited coherence, and some inelastic scattering occurs in the sample. Thankfully, ptychographic algorithms have been developed to allow for a mixture of states in the object and illuminating probe[53], and to estimate the incoherent scattering background present in the diffraction data[54]. Both of these developments were used here (see "Methods"), resulting in the probe composition in Supplementary Fig. 3. Nonetheless, maximising the coherence of the electron beam still greatly benefits the reliability and reduces computational time and complexity of the reconstruction algorithm. In practice, this is achieved by choosing an electron source with minimal energy spread and selecting an appropriate angular range of the electron emission. Thus, here we used a cold field-emission electron gun (CFEG) using a single-crystal tungsten emitter, which offers both a narrow energy spread and high brightness, which was further stabilised with a non-evaporable getter pump[55].

At the 20 keV beam energy used here, the inelastic and elastic scattering cross sections are greater than at more conventional TEM beam energies (≥ 60 keV), though as previously described, the elastic cross-section increases at a greater rate, leading to a potential for improved information coefficient in sufficiently thin samples[29]. However, despite the greater increase in elastic scattering, the inelastic scattering is still significant, particularly when the samples are thicker. As energy losses from plasmons and core-electron excitations are

relatively insensitive to the primary beam energy, blurring from chromatic aberration in the objective and projector lens arrangement increases with a reduced primary beam energy. This occurs as chromatic aberration scales in proportion to the ratio of the energy loss to the primary beam energy. For this reason, low-energy TEM instruments generally need chromatic aberration image correctors to resolve features in the Ångström regime[1].

A method of producing a model of the incoherent diffraction background, which includes some of this chromatic blurring that has proved successful in extreme ultraviolet ptychography, is to take the mean of the diffraction data and convolve it with a blurring function[54] (Supplementary Information, section 1). Here, we applied this method and found it advantageous to the reconstruction of the Au/MoS₂ data, lowering the reconstruction's error metric from 0.92 to 0.78. However, applying this method to the Au/aC data did not produce any improvement. We attribute this to the method more accurately representing the incoherent blur surrounding individual diffraction discs when the diversity in the diffraction data is lower. Thus, low diversity in the diffraction data, which usually presents a challenge for ptychographic reconstruction, is also seen here to present an opportunity for improved incoherent background estimation and modelling.

From a reconstruction of the sample exit-wave and its FT (Fig. 3a, b), we see features with a spatial frequency of at least 1.46 Å⁻¹, encircled in yellow, corresponding to the 0.683 Å spacing of the {401} plane group in the $MoS_2$. We note that our Ewald sphere radius at 20 keV is 11.6 Å⁻¹, so upon taking the c-direction unit cell dimension of $MoS_2$ as 12.3 Å[56], we enter the first order Laue zone for diffraction after the {300} family of planes. Adjacent to {401} peaks are further peaks at 1.60 Å⁻¹ (63 pm spacing), which may correspond to {533} planes of the ⟨111⟩ oriented Au islands or be related to Moiré effects arising from overlaying Au on $MoS_2$, which has very similar periodicity[52]. We also see (1/3){Au:422} (0.40 Å⁻¹) peaks along with other satellite peaks, which have until now only been seen in AC-TEM imaging of this material system[52].

We further make comparisons between the experimental reconstruction and a simulated reconstruction, created from diffraction data produced from a multi-slice model of the sample. Here, 5 layers of $MoS_2$ and an angle of 2.9° between the $MoS_2$ c-axis and the beam axis gave the best fit with the observed diffraction data, and the Au island was modelled with a tapered edge thickness (see Supplementary Information, section 4.1). If our experimental arrangement allowed us to realise an on-axis condition (see "Methods"), and plane wave illumination was provided, the exit wave in Fig. 4a, b would be expected. However, at our experimental sample orientation, we are not looking directly down atomic columns (see schematic in Fig. 4d), and adjacent atomic columns begin to overlap with each another in a depth-projected image. The result is that the plane wave illuminated exit-wave (Fig. 4c and Supplementary Fig. 9c) and simulated ptychographic reconstruction exit wave (Fig. 4e and Supplementary Fig. 9b) from the tilted sample model do not allow the smaller hexagonal motif in the on-axis case (Fig. 4b) to be clearly seen. The exit-wave phase-shift determined from ptychography (Fig. 4e) differed, as expected, from a plane-wave illuminated TEM image exit-wave (Fig. 4c). These differences were expected given the relatively low number of slices used in the reconstruction, as discussed in Methods.

Looking at enlarged views of the reconstruction (Fig. 3c, d, and Supplementary Fig. 9a), features directly related to an approximate 63 pm spacing can be directly identified. Similar features are also present in the simulated reconstructions (Supplementary Fig. 9b). Intensity profiles from experimental and model reconstructions (Fig. 3e and Supplementary Fig. 9d) appear to resolve lattice-fringe features, showing a mid-gap to peak intensity ratio of ~ 0.6. Although our models and reconstructions are in approximate agreement, the features present in the Moiré pattern produced from the Au on the $MoS_2$ do not represent on-axis uniformly separated single atomic

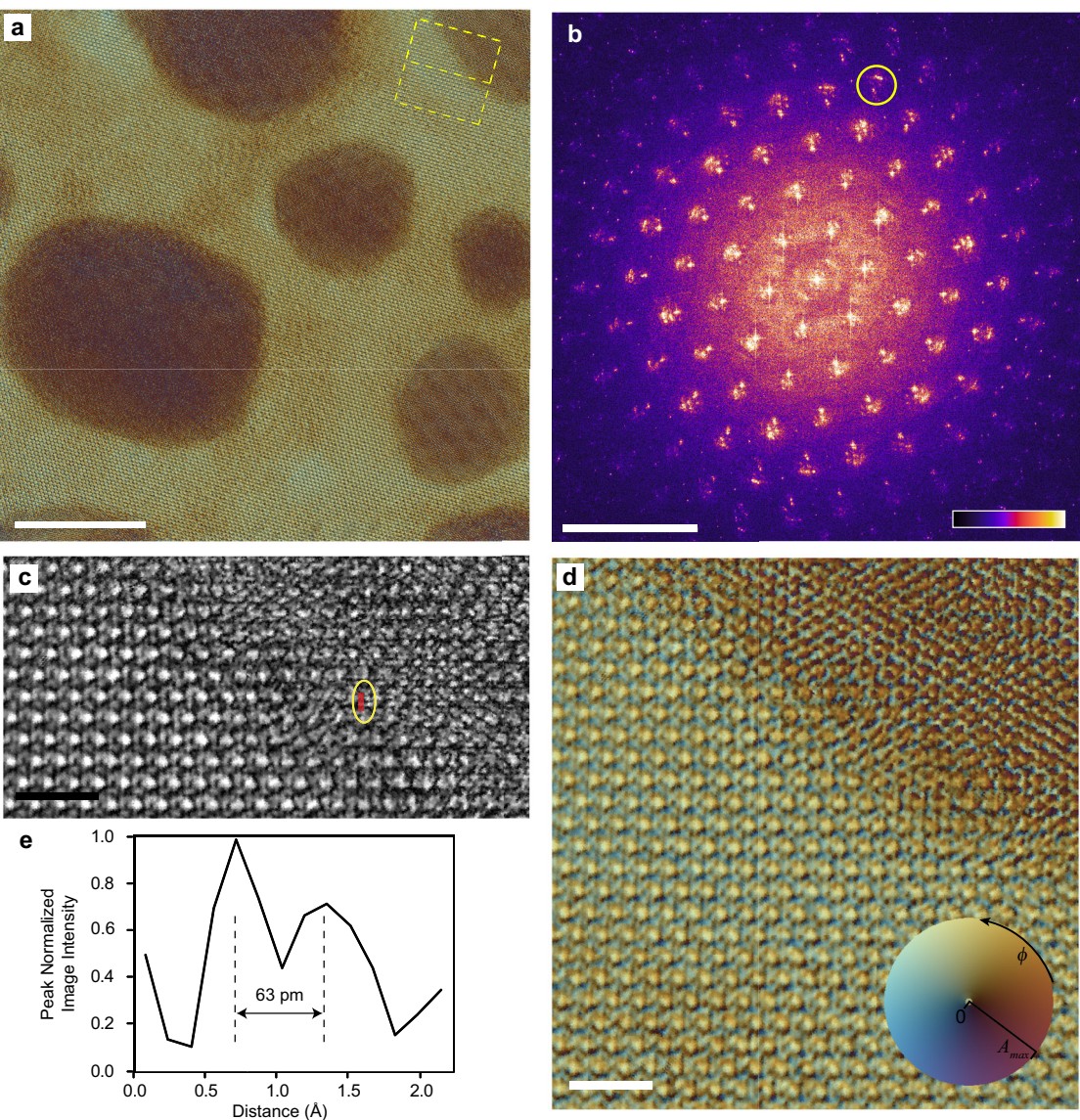

**Fig. 3 | Gold islands on MoS₂ reconstructions. a** Ptychographic reconstructed exit wave for 20 keV electrons transmitted through gold islands on few-layer MoS₂. **b** Amplitude of the Fourier transform (FT) of the intensity of the reconstructed exit wave (gamma adjusted to 0.9, upper and lower 1% of pixels saturated, with linear colour-scale at the lower right). **c** Amplitude and (**d**) phase-amplitude images of the region outlined in yellow in the upper right of (**a**). **e** Line profile of the reconstructed amplitude in the region marked with a red line and enclosed within the yellow ellipse in panel (**c**). Scalebars: **a** – 10 nm, **b** – 1 Å⁻¹, **c** – 1 nm, **d** – 1 nm. The phase and amplitude of the complex quantities in (**a** and **d**) are presented using a colour scale, which is inset in (**d**). Here $\phi$ represents the phase in radians, and $A_{max}$ is the maximum presented amplitude of the reconstructed images. The amplitude image presented in (**c**) was cropped from the full field of view amplitude image, where the upper and lower 1% of pixel values were saturated and linear grey-scale value mapping is used for the amplitude between 0.2 and $A_{max}$ = 4.1.

columns, given the tilt angle of the multiple layers in our sample (depicted in Fig. 4d). Even in the case of an on-axis sample, where aligned atomic columns act as the point-like objects to be resolved, a reliable resolution measure should include consideration of the signal-to-noise (SNR) ratio in the region, which is not trivial to determine[57]. Extracted line profiles (Fig. 3e and Supplementary Fig. 9d) are included to add support to the argument of there being useful repeatable information from independent reconstructions in the region of the maximum resolution (~67 pm) that is determined from the FRC method. Consequently, the resolution measure from the FRC method appears to us as the most useful and reliable resolution measure, as it involves multiple reconstructions and consideration of noise.

However, we recognise there are many other resolution measures in use in electron microscopy[49,58–60].

Within the context of the FRC resolution measure, our work shows the capability of ptychography using so-called dark-field scattered electrons to yield information in the reconstructions that is beyond the Abbe resolution limit. Simulations presented in Supplementary Fig. 8h–k further support this finding by showing that two isolated peak-like features, separated by around 67 pm can be resolved in an (infinite dose) reconstruction, when using a beam convergence angle of ≤10 mrad with a 20 keV electron beam, such as was used in our experiments. Future higher accuracy numerical and analytical work may reveal the variation and origins of any residual positional

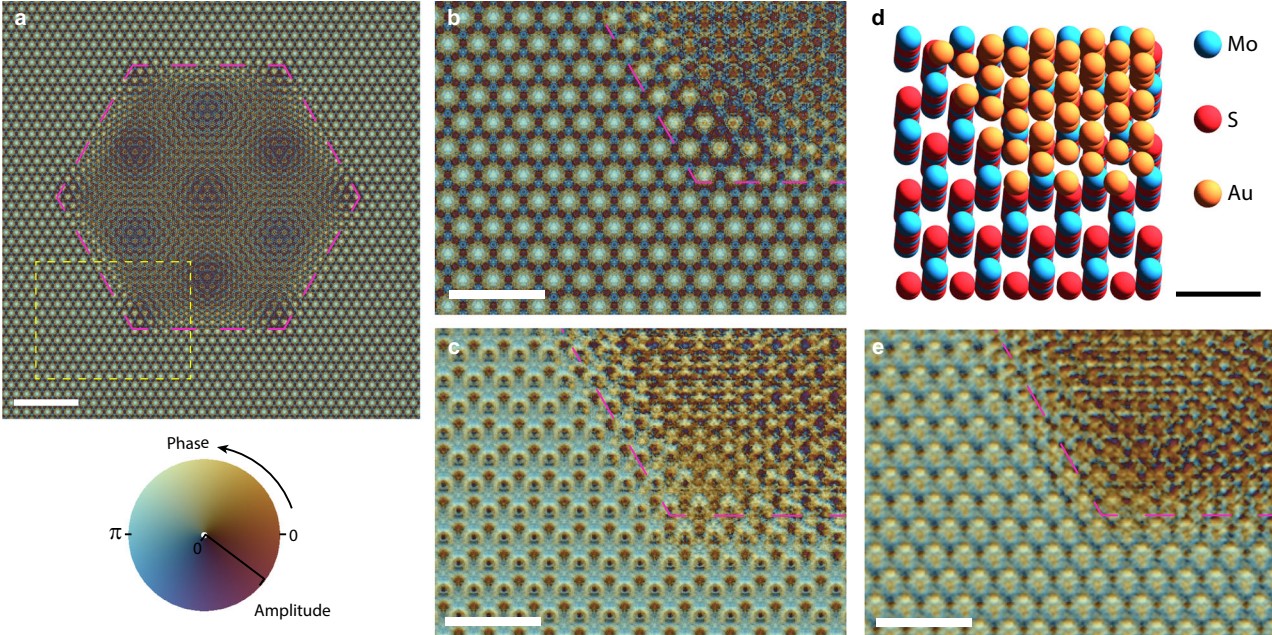

**Fig. 4 | Simulated Au/MoS2 images and reconstructions. a** Simulated exit wave for a gold island on MoS$_2$ using normally incident plane wave 20 keV illumination. **b** An enlarged view of the region indicated with the dashed yellow rectangle in (**a**) and (**c**), the plane wave illuminated exit wave from the same region when the sample is tilted at 2.9° from the MoS$_2$ *c*-axis, and relative to the electron beam illumination. **d** Visualisation of the atoms at the edge of the modelled gold island, viewed from the electron beam direction, where the sample is tilted at 2.9°. The gold island is modelled as having a frustrum geometrical from (see Supplementary Information) and thus has a tapered edge, seen by the gradually increasing number of gold atoms at the edge. **e** Exit-wave produced from a ptychographic reconstruction performed upon simulated diffraction data where the sample is tilted. The colour scale for the presented amplitude and phase in parts of (**a**–**c** and **e**) is shown in the lower left. Scalebars: (**a**) – 2 nm; (**b, c, e**) – 1 nm, (**d**) – 0.5 nm.

inaccuracy which is beyond, or in a similar range as, the approximate ± 10 pm registration accuracy between our simulated ptychographic reconstructions.

## Discussion

We experimentally show that a resolution measure of at least 67 pm can be achieved using ptychography in a non-aberration corrected scanning electron microscope, operating in a transmission mode with a 20 keV electron beam. This demonstrates a factor of 8.9 improvement upon the Abbe resolution, using $\alpha = 7.2$ mrad from the Au/aC reconstruction, and a resolution to electron wavelength ratio of 7.8. This ratio appears to go beyond that of earlier sub-Å ptychography demonstrations[61], albeit where the information we obtain here is within a less constrained multi-slice approximation, where potential phase reconstruction ambiguities may occur when using thicker samples. We achieve this resolution measure while being at ≤ 0.25 of the beam energy of these earlier sub-Ångström studies. These earlier studies gave an information limited effective resolution of 39 pm[21] at 80 keV and an effective Rayleigh resolution of 18pm[19] at 300 keV energy, respectively corresponding to resolution-to-wavelength ratios of 9.3 and 9.1. Achieving sub-Å resolution and lowering this ratio, while at a much lower beam energy in a non-aberration corrected SEM, increases the prospect of more widely accessible high-resolution electron microscopy.

As further detailed in Methods, we achieved this through using a CFEG SEM, an immersion objective lens, an additional single simple projector lens, and a low-energy optimised hybrid direct electron detector[62], along with developing a method to accurately determine and correct for distortion in collected diffraction data. We also ensured that the slice separation in our multi-slice reconstruction permitted valid reconstructions, and limited our sample thickness to the region where some improvement in the information coefficient compared to conventional TEM beam energies would be expected for lighter elements. Given the sample thickness constraints of lower

beam energies, we anticipate our advance to most immediately benefit research and production of 2D material-based devices[37], and facilitate in situ atomic scale investigations within the larger sample chamber of typical SEM instruments.

In the longer term, this may assist in the structural determination of proteins with masses below about 100 kDa, by circumventing the potentially prohibitive financial cost, space and personnel requirement of conventional higher beam-energy TEMs[2]. However, in the case of proteins, the dark-field signal arising from electrons scattered outside the brightfield disc is expected to be weak. At the low electron dose conditions required for non-destructively imaging proteins, advances in current ptychographic algorithms, likely fused with single particle analysis methods, would be required to make effective use of the weak dark field signal and to extend the resolution beyond the bright-field disk-limited Abbe resolution of $\lambda/2\alpha$. Consequently, though our work in its current form cannot directly result in atomic resolution protein images, it demonstrates some the potential of 20 keV electron ptychography to yield sub-Å information that may assist in future structural determination methods.

## Methods

For the Au/aC experiments, we used a combined gold and graphitised carbon on an amorphous carbon sample from Ted Pella Inc. (CA, USA), part number #638. This sample underwent approximately 2 min of cleaning in a Hitachi High-Tech ZoneTEM ozone – deep UV cleaning system at its lowest pressure, to remove residual organic contamination and to slightly thin the aC support film by ~ 2 nm. This gave an underlying aC thickness of ~ 15 nm, as measured by electron energy loss spectroscopy in a separate electron microscope at 200 keV, by analysing the ratio of zero-loss peak intensity to the total intensity. The preparation process for the Au/MoS$_2$ sample is described elsewhere[52]. However, here we determined that the MoS$_2$ was composed of 5 monolayers, which implies a thickness of 3.1 nm, using model-fitting between simulated and experimental convergent beam electron diffraction

patterns, obtained by 4D-STEM (see Supplementary Information, Section 4.1). The gold islands on the MoS$_2$ have diameters ranging from about 5 – 20 nm, with the thickness of a smaller island (top right of Fig. 3a) estimated to be approximately 4 nm through analysis of the phase of our ptychographic reconstruction and matching to simulation (see also Supplementary Information, Section 4.1).

Experiments were performed on a modified Hitachi SU9000 scanning electron microscope (SEM). This instrument can produce a maximum beam energy of 30 keV, though here it was used at 20 keV. This SEM is also equipped with a cold field-emission electron gun (CFEG) stabilised with a non-evaporable getter pump[55]. Samples are placed in the immersion-objective magnetic-lens gap (see Supplementary Fig. 10) using a conventional side-entry style transmission electron microscope (TEM) holder. For our work, we used a single tilt TEM sample holder. Compared to a double-tilt holder, this made achieving an on-axis condition impractical with the MoS$_2$ sample, particularly given that we aimed to minimise its time under electron beam irradiation to reduce possible beam damage[36] and minimise beam-induced build up of surface contamination. The instrument was modified from a standard design to include a small projector lens (item 6 in Supplementary Fig. 10a) beneath the objective lens. This additional lens improves control of the camera length (magnification of the diffraction plane), which is realised in combination with varying the objective lens current and the sample $z$-position (height) within the objective lens gap.

Beneath the retractable bright field detector of the SU9000, we added a hybrid-type pixel array direct electron detector. This detector is based on the EIGER detector design[62], as provided in a Quadro family camera from Dectris AG, Switzerland. For this work, the detector was customised for low-energy electron (6 – 30 keV) performance by removing the uppermost aluminium layer, thus eliminating a dead layer in the detector which becomes increasingly significant at low energies[62]. The detector collects 512-by-512 pixel-count images (where each pixel is 75 μm square) and can operate at a full-frame rate of 4500 Hz (2250 Hz) when operating at 8-bit (or 16-bit) pixel depth. However, for our experiments, we used a frame rate of 200 Hz, with no deadtime between frames, giving a 5 ms exposure for each collected diffraction pattern. At our chosen counting threshold, we determined a mean gain of 0.85 counts/e$^-$.

For the gold on amorphous carbon (Au/aC) data acquisition, our mean beam step size was 0.5 nm based on a square grid of 90 by 90 points, whereas for the Au/MoS$_2$ sample, a step size of 0.9 nm was used on a 45 by 45 grid. This relates to 45-by-45 nm and 40.5-by-40.5 nm fields of view, acquired in 40.5 and 10 seconds, respectively, for the Au/aC and Au/MoS$_2$ datasets. Pseudorandom, but known, offsets from this grid, no greater than ± 50% of the step size, were added to the otherwise regular grid. The mean effective overlap of the illumination regions determined from the probe reconstructions (Supplementary Figs. 3 and 4, and Supplementary Information, Section 2) were thus determined to be 0.875 and 0.74, given the effective illumination diameters from the probe reconstructions of approximately 4 nm and 3.5 nm for Au/aC and Au/MoS$_2$, respectively.

The electron beam current, measured using a pre-specimen insertable Faraday cup, was 85 pA for the Au/aC and 40 pA Au/MoS$_2$ experiments. Thus, given the fields of view, the mean electron dose received by these samples were thus $1.06 \times 10^5$ e$^-$/Å$^2$ and $1.56 \times 10^4$ e$^-$/Å$^2$, respectively. For the Au/aC data, the mean pixel count received on the Quadro detector was 5.035 counts/pix. For our 5 ms frame time, this gives an average rate of $2.64 \times 10^8$ counts/sec in total across the detector. With our detector giving 0.85 counts per electron at 20 keV, this relates to approximately 49.8 pA current on the detector, which is approximately 60 % of the incident current. A similar calculation on the Au/MoS$_2$ data indicates that 65% of the incident current was collected on the pixelated detector. The remainder would either be directed towards the HAADF detector (located above the pixelated detector

and having a larger angular range), backscattered from the sample, or did not contribute enough energy to an individual pixel to pass the counting threshold of the detector, which is especially likely in the case of inelastic scattering with the specimen.

Example collected diffraction patterns for the Au/aC and Au/MoS$_2$ datasets are given in Supplementary Fig. 1(a, c), respectively. During the setup and alignment, the projector alignment coils were adjusted to give a distortion that visually appeared to be radial, which appeared coincident with the bright-field disc remaining approximately stationary while the projector lens was varied. In addition, we aimed to position the bright-field disc away from the central sensor segment boundary, which forms a cross-like artifact on the Quadro detector. As ptychographic algorithms require the unscattered beam to be at the centre of the diffraction images, the data was digitally shifted post-acquisition to recenter the beam in the diffraction plane. Regions outside to the original diffraction data were set to a value of zero and are hatched in green in Supplementary Fig. 1. During reconstruction, these *mask* regions, which are outside the original field of collection and so do not represent real collected data, are left to float. The algorithm will not attempt to match the intensity in this extrapolated or set-to-zero region, or use this region in the evaluation of the error metric (see Supplementary Information).

A ptychographic reconstruction was first made on the as collected Au/aC diffraction data (see Supplementary Fig. 2a for process flow). A radial average of the FT of the reconstruction (Supplementary Fig. 2b, red line) was then fitted – using a pincushion distortion model as described later – to a powder diffraction pattern for gold (Supplementary Fig. 2b, orange line), to determine the distortion present in the collected diffraction data. The extracted distortion model was subsequently used to undistort the diffraction data, prior to performing another reconstruction. Following this, another comparison was made between the FT of the reconstruction and model diffraction data. If the model and reconstruction FT radial peak positions differed by more than 1%, another reconstruction and diffraction-model fit was performed, followed by a further distortion correction to the diffraction data. With our data, two reconstruction and fit cycles produced agreement between the peak positions of better than 0.5% out to the {135} diffraction ring of gold. Key parameters for the Au/aC reconstruction algorithm are given next, followed by a description of the diffraction data model fitting.

Here, the ptychographic reconstructions sequentially used the single-slice ePIE[26] algorithm, an iterative least-squares maximum likelihood (LSQ-ML) solver for ptychography[63] implemented within the PtychoShelves code[64], and a multi-slice adaptation of (MS-)LSQ-ML, which was implemented within a modified version of PtychoShelves[19]. The reconstructions for the Au/aC (Au/MoS$_2$) samples used 50 iterations of ePIE followed by 300 iterations of LSQ-ML, and 600 (1700) iterations of MS-LSQ-ML. The first 300 iterations of LS-LSQ-ML used 2 slices, and the remaining iterations employed 4 slices separated by 20 Å, combined with 3 incoherently summed probe modes[53] (see Supplementary Methods, and Supplementary Table 1). To gain successful reconstructions in the MoS$_2$ sample, avoiding crystalline sample related artefacts in the reconstructed electron probe, we found it necessary to constrain the probe's FT to be within an angular range less than the innermost extent of the first-order diffraction discs. Given our probe-illumination angle, this constraint was reasonable and physical, simply confining the beam to have its known circular aperture constrained form. A similar constraint on the FT of the probe was applied during the Au/aC dataset reconstruction. This was found to improve the physicality of the reconstructed probe by eliminating the presence of weak, randomly oriented lattice like fringes in the second probe mode that appeared in the probe reconstruction without this constraint. Though the presence of these fringes in the probe does not alter the FRC resolution measure limit, it reduced the total area under the FRC characteristic in Fig. 2 and reduces the perceived clarity of

fringes in the reconstruction (see Supplementary Fig. 11). In addition, for the Au/aC reconstruction, we investigated using an initial object guess that had a flat (constant) phase and amplitude, and an initial object, $O_i$, with a constant real component and a small random imaginary component, where $\max(\mathrm{Im}(O_i))/\mathrm{Re}(O_i) = 10^{-6}$. This random initial object also produced a small improvement to the FRC characteristic. For the Au/$MoS_2$ reconstruction, we used a constant amplitude and phase initial object. Though a random object here may have improved this reconstruction too, we did not investigate it further, as here we had insufficient data to perform an FRC analysis.

To determine the distortion in the FT of the Au/aC reconstruction, we first determined a radial average of the FT's squared magnitude (see Supplementary Fig. 2b). A distortion mapped ideal radial diffraction profile for randomly oriented gold powder, calculated using Crystal-Maker (from CrystalMaker Software Limited, Oxfordshire, U.K.) with a 20 keV beam energy, was then fitted to this average profile by adjusting the overall scale or magnification, $M$, of this ideal diffraction intensity profile and introducing a pincushion distortion coefficient, $k$, defined in the relationship,

$$s = Mr(1 + kr^2) \tag{1}$$

where $s$ and $r$ are the distorted and undistorted radial distances (in reciprocal space) of the diffraction intensity profile respectively. Here, the fitting was performed by minimising the Euclidean distance between the experimental and model radial profiles, using the simulated annealing algorithm, though we also found that genetic algorithms and pattern search methods yielded very similar results. We also manually identify peaks in the radial FT, and compare the measured radial frequency, $s$, of the peaks with the expected radial frequency $r$, using a ($s/r$) vs. $r^2$ plot as in Supplementary Fig. 2c, where $M$ and $k$ can then be obtained from the intercept and slope. A linear fit is shown in Supplementary Fig. 2c, indicating the appropriateness of the pincushion distortion model. In a magnetic lens system, we expect some azimuthal (spiral-like) rotation distortion of the diffraction data that also varies with the same polynomial order as the form in Eq. 1 in the radial coordinate[41]. Normally, the azimuthal distortion is much less significant than the radial part[40], which was the case here. Examining the undistorted $MoS_2$ diffraction data (Supplementary Fig. 1(c, d)), the diffraction discs are aligned along straight lines, and there is not a measurable spiral-like distortion.

The pincushion distortion depicted in Supplementary Fig. 2d, e, was corrected in our diffraction patterns using the distortion parameter, $k$, determined from an earlier Au/aC reconstruction. The magnification correction, given through $M$, was simply corrected for by adjusting the reconstructed pixel size input to the reconstruction algorithm, rather scaling the image. In the distortion correction, we set $M = 1$. After distortion correction, the reconstructed pixel size for the Au/aC and Au/$MoS_2$ datasets were 23.8 pm and 33.14 pm, respectively. In the final 1000 iterations of the Au/$MoS_2$ reconstruction, the diffraction data was padded by a factor of 2, resulting in a reconstructed pixel size of 16.57 pm using a super-resolution type approach[65] (see Supplementary Information). Example corrected diffraction patterns from these datasets are shown in Supplementary Fig. 2b, d.

The most widely used image distortion correction routines effectively assume that pixel values represent delta function sampling of the image intensity, and consider where this point value must be mapped to. However, here the values of our diffraction data pixels represent the integrated diffracted wave intensity over a finite-sized (75 μm square) square pixels. Thus, in the same manner as distortion correction has been applied to fMRI imaging[66] we adjusted the intensity of the pixel values in our distortion correction algorithm to take account of the compression (or, in general, expansion) of image patches between the distorted and undistorted spaces. This adjustment is generally not considered when correcting for general-purpose image

distortions, including for crystallographic purposes[41]. However, for ptychography, intensity adjustment is important: without it, electron counts on the undistorted pattern would be modified in a non-linear manner, potentially corrupting the centre of mass of our diffraction discs, as described in the Supplementary Information, Section 1.2.

## Data availability

The raw and distortion corrected experimental diffraction data, synthetic diffraction data, and all the reconstruction outputs presented in this study, along with some additional outputs from intermediate reconstruction have been deposited in the Canadian Federated Research Data Repository[67], under accession code https://doi.org/10.20383/103.01357.

## Code availability

The core PtychoShelves package used for the reconstructions was developed by the Science IT and the coherent X-ray scattering (CXS) groups, Paul Scherrer Institute, Switzerland and is available at https://www.psi.ch/en/sls/csaxs/software. The multi-slice adaptation of PtychoShelves for electron microscopy used here, called PtychoShelves_EM, was developed by members of Cornell University, et al., and is available at https://doi.org/10.5281/zenodo.4659690. Code developed at the University of Victoria, to wrap PtychoShelves_EM, include the aperture constraint, and provide the analysis performed here, is freely available at https://github.com/ArthurBlackburn/PtychoRunner, or from a Zenodo archive[68]. Prismatic software, used for producing simulated data, was developed by members of Lawrence Berkeley National Laboratory, et al. (see SI section 4), and is available at https://prism-em.com/. Our wrapper code for operating Prismatic for ptychographic data simulation is available at https://gitlab.com/mrfitzpa/prismatique.

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

## Acknowledgements

We thank Kate Reidy and Frances Ross (MIT) for providing the Au/MoS$_2$ sample. The extensive support given from Dectris AG (Switzerland) and their provision and development of the Quadro camera that expedited this work is gratefully acknowledged. This work was part-funded by the Natural Sciences and Engineering Research Council of Canada (NSERC) from a Collaborative Research and Development grant (CRDPJ 543431, obtained by A.B. and used to support C.C. and M.R.F.) partnering with Hitachi High-Tech Canada. Part of the data processing was carried out using the cSAXS ptychography MATLAB package developed by the Science IT and the coherent X-ray scattering (CXS) groups, Paul Scherrer Institute, Switzerland. These codes and supporting simulations were performed using the Advanced Research Computing (ARC) facilities of Digital Research Alliance of Canada. Members of Hitachi High-Tech, Naka, Japan, who developed and integrated the projector lens on the experimental SU9000 instrument, are thanked and acknowledged, with special thanks going to Kazutoshi Kaji, Satoshi Okada, and Toshi Agemura.

## Author contributions

A.B. conceived the experiments, distortion correction method, collected data on Au/aC, modified PtychoShelves code to implement new constraints, including wrappers to perform reconstructions with a wide range of parameters and probe models, and developed code to extract aberrations from reconstructed probe. C. C. collected data on Au/MoS$_2$ and Au/aC and performed some initial reconstructions. M.F. created simulated data on MoS$_2$, through developing wrappers around Prismatic software. R.M. created the 4D-STEM acquisition system and integrated the Dectris Quadro detector. A.B. draughted the manuscript, and all authors reviewed the entire manuscript and figures, and contributed to respective sections according to their roles above.

## Competing interests

Hitachi High-Tech provided the instrument used in this study, supports a research chair for AMB, and employs RAM. The remaining authors declare no competing interests.
