## [Transparent Peer Review file · Nature Communications]

Sub-ångström resolution ptychography in a scanning electron microscope at 20 keV

Corresponding Author: Professor Arthur Blackburn

Version 0:

Reviewer comments:

Reviewer #1

(Remarks to the Author)

The authors experimentally demonstrate high-resolution dark-field multi-slice ptychography at 20kV in a modified SEM. The result is very relevant for 2D materials research and electron microscopy technique development, which has wide applications in materials and biological sciences.

The paper is well written and should be published in NatComm, after one major and a few minor concerns are addressed.

The major concern is that the reviewer thinks that there is a clear distinction between dark-field ptychography as practiced in this paper, and "standard" ptychography in the way they transfer spatial frequencies and allow for below unit-cell position determination.

In the paper, the authors write:

"Note that at spatial frequencies not originating from the gold particles, namely the amorphous carbon support film, we expected only a relatively low or insignificant correlation between the reconstructed images. Indeed, thin amorphous carbon support films are generally chosen in TEM studies for their absence of readily discernible structure that would otherwise confound imaging of the supported material of interest in TEM images. Thus, note that the low FRC measure between the diffraction peaks is attributed to the sample and is not a deficiency in the algorithms to reconstruct at these spatial frequencies. "

In dark-field ptychography, phase information is only encoded in the interference between Bragg-disks, and in the relatively few pixels of the bright-field disk.

The reviewer thinks this leads to gaps in the spatial frequency transfer of dark-field ptychography. Some spatial frequencies never receive coherently interfering electrons and therefore the phase information of these frequencies cannot be uniquely determined.

This results for example in the fact that atomically sharp interfaces cannot be reconstructed with dark-field ptychography, as seen in the simulation in Fig 4. Some atomic lattice from the gold is leaking into the MoS₂ area, although we have perfect, simulated data.

Also in the Au/aC image, we do not see atomically sharp interfaces, and we can make out edges of gold particles where the lattice is leaking into the aC area.

The following could be done to illustrate or refute this point: Instead of the FRC, which just compares two reconstructions without ground truth, the authors should plot the spatial-frequency resolved SNR(q) (eq 44 in <https://iopscience.iop.org/article/10.1088/1367-2630/14/6/063004>). For Fig. 4, ground truth data is available, so the SNR can be calculated. Also, the sample contains a sharp edge, so that all spatial frequencies should be present in the ground truth.

As comparison, perform a reconstruction of the same sample in Fig. 4 with the convergence angle chosen such that the first diffraction order disc and bright field disc have some overlap and compute the SNR(q). Can this "standard" ptychography setup reconstruct spatial frequencies which the dark-field approach cannot reconstruct?

I think this point is important and can be evaluated with existing code and infrastructure, thus it should be not too much extra work. Discussion of the results should improve the understanding of this optical configuration of ptychography.

Another point is the accurate position determination. The reviewer thinks that uni-cell hops cannot be resolved in dark-field ptychography, which may also be another cause of lattice delocalization, like in Ref 33. If the authors think that they can reliably resolve unit-cell hops with their algorithm, they should show it with simulations. If not, they should point it out as future research to the reader.

Minor points:

1) since the setup is non-standard, a Ray diagram of the optics would help the reader understand configuration

2) line 15 "Furthermore, lower electron beam energies are expected to give an increased information coefficient, provided the specimen is commensurately thinner and composed mainly of light elements."

The reader cannot be expected to know the paper that defines the term "information coefficient", please replace with standard, known terms

3) line 27 "and in the longer-term assist in the structural determination of proteins with masses below about 100 kDa ..."

It should be mentioned that a different optical configuration with much larger convergence angle is necessary than presented in this study, since proteins are weakly scattering and produce much less dark-field signal. The resolution will be limited to a little more than $2 \times (\text{convergence angle})$

4) line 76 "Reprocessing the data from these early experiments using an algorithm to account for the non-ideal 77 camera response yielded some phase reconstructions which perhaps indicated the 022 planes (with 1.44 Å separation) of gold particles, though this was not confirmed using standard resolution assessment methods22."

The feature delocalization could be mentioned here, as visible also in Fig. 1,3, and 4 of this paper.

5) line 63 "Also, it was recently applied to electron microscopy of few layer TMDCs (e.g. MoS₂, MoSe₂, WS₂) and thicker samples of PrScO₃ 17-19 to give sub-Å information with a 80 keV¹⁹ and 300 keV^{17 65} beam energy, where it provided a 3 – 4x improvement over the conventional resolution of 66 the electron microscopes used in the experiments."

Recent publications making use of the depth-sectioning capabilities of MS-ptycho should be included, e.g.

18 H. Sha, Y. Ma, G. Cao, J. Cui, W. Yang, Q. Li and R. Yu, Nature Communications, 2023, 14, 162.

19 H. Sha, J. Cui and R. Yu, Science Advances, 2022, 8, eabn2275.

20 H. Zhang, G. Li, J. Zhang, D. Zhang, Z. Chen, X. Liu, P. Guo, Y. Zhu, C. Chen, L. Liu et al., Science, 2023, 380, 633–638.

6) Extended Data Figure 7: why are regions excluded?

"To eliminate the contribution to our resolution measure of subregions that had little or no useful information (such as regions containing mainly amorphous carbon), regions with a mean FRC below a threshold given by Otsu's method were excluded from further consideration."

Other authors use amorphous carbon to get CTF in standard ptychography. The reviewer thinks this may be a result of the gaps in spatial frequency transfer. If this is so, it should be pointed out to the reader, see main concern.

7) "Defocused probe electron 188 ptychography, as used here, does not require a high-resolution, finely focused, or spherically aberration corrected 189 electron probe to achieve a high-resolution reconstruction. "

Again, please discern between dark-field and normal ptychography.

8) it should be mentioned that the dark-field optical configuration as used here will not result in atomic resolution protein images

9) what is the expected depth resolution of dark-field MS-ptycho? has this been evaluated by you with simulations? if not, point it out as future research

10) "We experimentally show that a resolution of at least 67 pm can be achieved using ptychography in a non266 aberration corrected scanning electron microscope, operating in a transmission mode with a 20 keV electron 267 beam"

Does it provide continuous transfer of spatial frequencies like standard ptychography? See main concern.

Reviewer #2

(Remarks to the Author)

This paper reports an important advance in low-energy transmission electron microscopy (TEM). With the advent of greatly improved electron lenses, which use multipole fields to correct for the aberrations inherent in conventional round magnetic lenses, the resolution of electron microscopes has quickly improved over the last 20 years. Given these enhanced lenses, a major thrust of aberration correction is to lower the electron voltage from the typical TEM acceleration of 200keV down to 60keV or lower. At lower energy, the electrons have longer wavelength, and can only provide intrinsically lower resolution: aberration correction can compensate for this. The rationale of low keV is to reduce damage events in the specimen (although some types of damage can be increased at lower keV). This strategy is sometime called 'soft microscopy' and promises to open up lots of new science, especially the understanding of structures consisting of only a few atomic layers.

However, the low-energy aberration-correction approach has met with further difficulties relating to decoherence due to thermal magnetic field noise in the electron optics. It also requires very complex and expensive instrumentation.

This paper reports on a different approach to very low energy transmission microscopy: the application of ptychography within a conventional scanning electron microscope (SEM), which is used with 20keV electrons (extremely low for transmission microscopy). Ptychography is very well established in X-ray microscopy. It utilises diffraction patterns that do not need to go through a lens. A proof of principle of ptychography in SEM was demonstrated by Humphry et. al. in 2012 but I am not aware of any further work on this technique since then.

The headline result is that the authors' microscope obtains a resolution of 67pm. Given the extreme low energy of the electrons involved, this is an extraordinarily impressive result, which could deliver all sorts of new science. As the authors also observe, an SEM is a fraction of the cost of a conventional aberration corrected machine and has a much smaller laboratory footprint.

However, there are some improvements in the paper which I would like to see implemented, and which I describe in detail below.

1) I want to start with what I think is its biggest weakness: the measurement of the resolution that it has purportedly been achieved. Leaving aside for now the diffractograms in Figure 1c, this headline result depends upon the line plot in Figure 3e, which itself depends on Figure 3c. Looking in detail at Figure 3c we can identify three distinct areas. Let us call the area with the oval marker, area 'A'. At the top of the figure, at about 15 degrees right from due North from the oval, there is a hexagonal crystalline area of a few unit cells, call it 'B'. At about 80 degrees from due North from the oval, and about halfway to the edge of the image, there is another crystalline area, call it 'C'. Area B appears to have two to three adjacent crystalline hexagonal unit cells of the same shape/size as the underlying MoS₂ at the bottom left of the simulation in Figure 4b. Area C is also crystalline, but with a much smaller unit cell, possibly also hexagonal: structures like this do not seem to be visible in Figure 4e, but are seen in Figure 4b (the untilted simulation). Meanwhile area A, which is used to confirm the real-space observation of the resolution record, is semi-amorphous, or at least very confused (with neither structures B or C prominent).

What is going on here? We know the two layers are incommensurate, giving rise to the Moire pattern in Figure 4a simulation, but the scale of the variation of the fringes is much larger, I think, than the field of view in Figs 3c/d, and so it seems unlikely that the Moire effect can transition so quickly from B to C. The only theoretical calculation of relevance is Fig 4e – the ptychographic reconstruction that should correspond to the experimental images figs 3c/d. It would help if we could be shown the intensity of Figure 4e, in order to make a direct comparison to Fig 3c, from which the actual line plot proving the resolution has been taken.

On a different tack, it is argued that the diffractogram in Fig 1c, and the corresponding ring correlation in Fig 2, proves the 67pm spacing is present in the image. It is true diffractogram analysis is often used to prove the resolution capability of a conventional TEM, but then the image is generated by an 'analogue' process: the image is formed directly by a lens. Here, the image is a reconstruction generated by applying a minimisation algorithm to very indirect data, which is never 100% reliable. There is therefore the possibility that the spacing is an artefact of the reconstruction. If this separation is indeed present in the real space image, the relevant fringes may not be properly localised at their real position (an effect that also happens in lens-based TEM – the presence of fringes of a given spacing is not the same as usable resolution at that spacing).

I think the authors should unpick these issues, because as it stands the structures visible in the relevant images are neither fully described or explained, and hence the resolution record claimed is not very secure.

2) The details of the projector lens are not described. First, I think it would help the reader if the relationships between probe size, sampling in the diffraction pattern, total size of the diffraction pattern and resolution were outlined. What range of these parameters can be reached with the projector lens? (10 degrees is mentioned – does it have to be that big? How far is the projector from the objective? – I presume there must be a cross-over below it. I personally would like to see a ray diagram, if only in the supplementary information. This is, after all, a key innovation in the experiment and the entire technique.

3) Does the large probe size help with reducing beam damage? The total flux/area through specimen will remain constant, but it has been proposed that scanning randomly could reduce time-dependent damage effects, or even local heating effects.

4) Why was the circular aperture constraint used during the reconstruction needed for one sample but not the other?

5) Shifting the probe can induce position-specific aberrations (coma, etc). Was there any evidence of this?

6) "We found that despite the diffraction data being distorted, a ptychographic reconstruction could be made, and its fast Fourier transform (FFT) yielded well-defined low-order diffraction peaks concentrated in rings, as expected for a polycrystalline sample." I note that in the case of putting a detector very close to the object, so that it is in the Fresnel condition, the ptychographic reconstruction comes out near-perfectly, even if a Fourier propagator is assumed in the algorithm. The difference is that the probe reconstruction develops a phase curvature reminiscent of a lens. This is because a pre-factor of a parabolic phase can be used to form the Fresnel integral, while still using the Fourier transform. The extra lens phase brings the parallel beams of the FT to a focus on the nearby detector. I say this because it is possible, I think, that the pin-cushion distortion may also be accounted for by a distorted reconstruction probe phase. If that's the case, the authors could avoid all the complexity of their pin-cushion mapping algorithm. No alteration to the paper required, just an idea for further work...

7) Somewhere in the paper I think the disadvantages of using a large probe should be mentioned, i.e. the loss of the analytical X-ray signal. Elemental determination is one of the biggest applications of SEM.

8) Caption fig 4d: There must be a better of saying 'Depiction of the simulated atomic positions simulated,'

9) Figure 1 has no 1d, although 'd' is in the figure caption.

There is no doubt that this paper makes an important contribution to the state-of-the-art of electron microscopy. It is technically sound and well written. I would strongly support its publication in Nature Communications. My only concern is the question of having proved conclusively the resolution that they claim has really been reached. The resolution is undoubtedly greatly increased over previous work, but is it by as much as the authors claim? More careful interpretation and discussion of the results is required, I think.

John Rodenburg

Reviewer #3

(Remarks to the Author)

This is an interesting paper that reports Ptychography at low voltage. The paper describes several applications which demonstrate enhanced resolution at 20kV compared to other imaging methods and provides some details of the instrumental and computational requirements used to achieve this.

As a technical improvement this is worth publishing in a technical journal. However, in my view it does not represent the substantial improvement that is needed for the wider readership of Nature Communications.

I also have a number of specific comments / criticisms of the current paper.

1. Most importantly the work reported is very similar to the recently reported work of Allen et al. (APL, 2023) who showed that ptychographic reconstructions beyond a 2 alpha limit could be achieved for MoS₂ samples at 30kV. The authors reference older work by Rodenburg et al using an SEM but do not consider this more recent work which provides a more general and efficient method for ensuring convergence than that described on page 8.

2. Throughout the paper there are unfortunate comparisons between the work reported and alternative imaging methods without reference to the differences between these. For example the resolution in an SEM is quoted as 5Å but this is for SE imaging which is not comparable to the ptychographic phase reported here.

3. A large part of the paper is devoted to methods for measuring and correcting the diffraction patterns used. This is simply a feature of the relatively simple post specimen optics employed here and would not be a problem for more conventional multi lens projective systems which easily attain distortion free data at 99.9%. This discussion might however be useful in the context of a paper describing a low cost instrument for ptychography.

4. There are many examples where comparisons / descriptions are only qualitative e.g. on p7 the reconstruction is described as stronger and sharper with no values given. Similarly on p3 the statement that "ptychography requires the illumination to have significant coherence..." could easily be quantified.

5. The description of a lack of FRC data (p7) between the Bragg reflections is confusing – amorphous carbon does have structure and the FRC approach should have a positive value as the sample is the same for the two subsets of data correlated.

6. There are a number of clear typographical errors; more than I would have expected. For example p8 incorrectly describes inelastic and not elastic propagation and p9 quotes a 68.3 Å spacing. Throughout the term FFT is used to describe the power spectra (or spectral densities) shown. This is incorrect the FFT is a computational approach (due to Cooley and Tukey) for calculating the discrete Fourier Transform of an object. Figure 1 a claims to show the exit wave but the data shown is real (I assume it is the phase) not complex as expected for the exit wave. The same applies to Figure 3 a. Finally extended data Figure 8 does not directly add scientific value to the work presented and should be deleted.

7. P10 contains a description of the sample mis tilt which could have been corrected making the comparisons between simulations and experiment more meaningful. I accept that only a single tilt holder was available but given that the instrument used has a conventional side entry goniometer, surely using a commercially available double tilt holder would have been possible.

8. On p10 there is match between experiment and simulations that is used to obtain a sample thickness to high precision. However the Stobbs factor is not considered.

9. The manuscript contains an excessive use of superlatives – “new records”, “dramatic” etc. These are not needed.

Overall if the above were addressed this work could be published in a specialist journal with, perhaps more emphasis on the instrumental modifications.

Version 1:

Reviewer comments:

Reviewer #1

(Remarks to the Author)

In Revision 1 the manuscript has improved significantly and many comments were resolved.

I acknowledge that the matter of information transfer in conjunction with multislice reconstruction requires more work to fully unravel, and the authors have made some strides and the necessary comments to point out the intricacies.

There are some remaining issues which should be resolved before publication - some slipped through the first round of the manuscript, some newly introduced in Rev1.

1) FRC depends on initialization - which initialization was used for the algorithm? If the same initialization was used for both independent reconstructions, this can lead to a biased FRC, as shown in [Allars, Frederick, et al. "Efficient large field of view electron phase imaging using near-field electron ptychography with a diffuser." *Ultramicroscopy* 231 (2021): 113257.]

2) Extended Fig. 3 shows lattice features in the probe mode #2, while Ext. Fig. 4 which used additional probe constraints does not show these lattice features. The mode is also quite strong with 19.2%. Did a probe constraint on the Au/aC sample not improve the reconstruction?

3) The new Extended Data Fig. 8: Yes, the figures h-j show that this spatial frequency can be resolved, but they show also exactly my concern: the atomic peaks in the 5mrad reconstruction are shifted with respect to the Ground Truth. The peaks in the 20mrad reconstruction match the ground truth.

The authors write in their rebuttal: ". Though this does not fully address the question posed, it gives some indication that when there are many spatial features present in the object (and so the object has a rich spatial diversity), accurate localization of isolated features appears possible."

Given Extended Fig. 8 I would object and say it is highly doubtful that isolated features can be localized accurately.

The text does not mention the fluence used to produce Extended Data Fig. 8 - was the fluence for the 20mrad and the 5mrad case the same and what was the fluence? The 20mrad data accurately localizes the atomic features while the 5mrad data does not.

Maybe one could show that the position of the isolated atoms is very weakly encoded in the 5mrad diffraction pattern and stronger encoded in the 20mrad diffraction pattern by comparing diffraction patterns from probe positions that are a few Angstroms apart. The 20mrad patterns should change much more strongly for small position changes.

4) VERY IMPORTANT: The code for the main result, i.e. the experimental reconstruction, is currently not included in the paper and just "Available on request". This is completely unacceptable for an algorithm-focused paper. The code to reproduce the experimental reconstructions must be shared openly.

5) Why did the authors opt to show only exit waves of the final model instead of the summed phase images of the final model, as now customary in multi-slice ptychography reconstruction, e.g. the original paper by Chen 2021 and the new ref 20-22? Some amplitude images are shown, but no phase images. Does phase unwrapping cause problems?

(Remarks on code availability)

Currently only the code for the simulations is shared.

The code for producing the main experimental result must be shared openly and not only "on request".

Reviewer #2

(Remarks to the Author)

The authors have addressed most of the issues raised by the referees. The paper is now rather long by the standards of

printed articles, but a brief look at Nature Comms suggests that this is quite normal. Some questions have not been answered fully, but only because to do so would require a very substantial additions to paper. In such instances, the authors have added to the text to highlight these complications. By leaving questions open, the paper invites others to think about, and work on, these matters, which is a good thing. (As an example - the effect on the fidelity of the reconstruction in the presence of a weakly scattering amorphous support overlaying a dominant periodic feature.) But it would over-burden the present work to cover all these nuances – it would also require a lot more work beyond the main scope of the paper. I think it is thorough enough as it stands.

(Remarks on code availability)

Reviewer #3

(Remarks to the Author)

The authors have clearly made significant efforts to address the comments raised in my review of their initial submission. Having read their responses to all of the referees comments I am now happy that the manuscript can be published. My only remaining minor point is that I feel that basing the use of an uncorrected SEM for Ptychography on cost grounds only tells part of the story - the availability of a large volume around the sample which could be used to introduce external stimuli more conveniently and flexibly than in a TEM is surely equally important.

(Remarks on code availability)

RESPONSE TO REVIEWERS COMMENTS

In this document we address the comments from the referees.

We have copied the referees' comments from the e-mail and placed them rightmost and in black text in the pages that follow.

Our responses are indented and coloured blue.

Portions which have been copied from the main text are not indented but are marked with bar on the left-hand side as shown here. Material which is added to the main text is copied here and shown underlined in red. For the sake of brevity not all minor deletions and edits from the main text are shown, but larger deletions are ~~shown by strikethrough.~~ We recommend the marked-up and 'clean' revised manuscripts provided are read in conjunction with this response.

Reviewer #1 (Remarks to the Author)

The authors experimentally demonstrate high-resolution dark-field multi-slice ptychography at 20kV in a modified SEM. The result is very relevant for 2D materials research and electron microscopy technique development, which has wide applications in materials and biological sciences.

The paper is well written and should be published in NatComm, after one major and a few minor concerns are addressed.

The major concern is that the reviewer thinks that there is a clear distinction between dark-field ptychography as practiced in this paper, and "standard" ptychography in the way they transfer spatial frequencies and allow for below unit-cell position determination.

In the paper, the authors write:

"Note that at spatial frequencies not originating from the gold particles, namely the amorphous carbon support film, we expected only a relatively low or insignificant correlation between the reconstructed images. Indeed, thin amorphous carbon support films are generally chosen in TEM studies for their absence of readily discernible structure that would otherwise confound imaging of the supported material of interest in TEM images. Thus, note that the low FRC measure between the diffraction peaks is attributed to the sample and is not a deficiency in the algorithms to reconstruct at these spatial frequencies. "

In dark-field ptychography, phase information is only encoded in the interference between Bragg-disks, and in the relatively few pixels of the bright-field disk.

The reviewer thinks this leads to gaps in the spatial frequency transfer of dark-field ptychography. Some spatial frequencies never receive coherently interfering electrons and therefore the phase information of these frequencies cannot be uniquely determined.

When dealing with a perfect crystal, free of any support or contaminating material, diffraction of course occurs only into the Bragg disks, determined by dynamical diffraction theory. Thus, yes, if there are regions of no overlap and no intensity in certain of diffraction data it would be expected that there would be regions of missing information and phase ambiguity in any reconstructed model of the specimen. However, here our samples (and many real samples) are not simply perfect crystals: the samples are either supported on amorphous carbon supports, or are perhaps undesirably coated with some likely hydro-carbon based contaminating material. This support film or contamination material, which is generally considered as amorphous, introduces additional structure in the sample, which has a wide range of spatial frequencies. In fact, the range of spatial frequencies present in amorphous materials is so wide that such materials are regularly used as a means of determining or inferring the phase contrast transfer function of electron microscopes.

The role of largely amorphous contamination layers in ptychography of otherwise crystalline samples has not yet been fully studied, but would be an interesting topic for future studies. We posit that these layers (and support films layers), through their wide range of spatial frequencies, and by virtue of multiple-electron scattering in the sample, may explain the consistency of the reconstructed phase between independent datasets that we see in our reconstructions of gold of amorphous carbon, at a resolution beyond that expected from simple consideration of the illumination beam half angle. In the case of overlying amorphous material, multiple scattering would mean that underlying crystalline sample layers are subjected to a much broader angular range of illumination than would be expected from a simple model of the beam that assumes no scattering before the crystalline sample is reached. In the case of the support film under the sample, multiple scattering gives a finite probability to there being features in the diffraction data that are

the result of beams diffracted from the crystalline specimen interfering with each other after scattering through the amorphous carbon. These patterns would thus allow ptychography to give some information on the relative phase of the crystalline diffracted beams, even if the apparent Bragg diffracted beams are non-overlapping.

Looking at it another way, one could imagine the 'Bragg disks' from the wide range of spatial frequencies in amorphous carbon support films (or contamination layers) to provide additional diffraction discs that overlap and thus tie together the otherwise separated (non-overlapping) Bragg discs of the crystalline material. When we say 'Bragg disks' from amorphous carbon, these disks are more usually referred to as a 'speckle pattern' in amorphous materials, which appear either away from (or coherently interfering with) the central brightfield disk depending on the convergence angle of the electron illumination, and the spatial frequency of structural features.

Thus here, we adopt the remedy of pointing out this possibility to the reader. We make it clearer that our samples are not the high-quality and very clean crystals that appear to have been used in earlier demonstration of ptychography on largely monocrystalline samples. We also note that this mechanism is not fully explored here and would be worthy further investigation in future work. (Note also later we have now included some simulations that lend credibility to being able to resolve features and reconstruct phase reliably at frequencies beyond the bright-field disc limited resolution, when no obvious overlapping brightfield discs are present).

Note the following modifications and additions on page 8:

In Fig. 2 we additionally plot the average radial profile of the FT of the reconstructed intensity and the diffraction intensity profile expected from randomly oriented gold particles, where we label the dominant Au diffraction peak indices. The non-crystalline structure of the amorphous carbon, where the structural information is broadly spread over a wide spatial frequency range, would require a greater electron dose to reliably determine its structure and show correlation between independent scans in comparison to crystalline gold which concentrates most of its scattered intensity into or near discrete peaks (or Bragg discs) that are easier to distinguish in the presence of inevitable dose-limited shot noise. Furthermore, the high electron beam intensity in a CFEG STEM system can readily induce structural rearrangement and instability in aC, through mechanisms such as electron beam sputtering and deposition⁴⁸, which further reduces the likelihood of correlations between reconstructions of amorphous carbon. These factors make a stable reconstruction of the aC much less likely than the gold particles, which in comparison remain stable in the electron beam. Thus, the low FRC measured at spatial frequencies between those of the Au diffraction peaks here is not necessarily a deficiency in the algorithms to reconstruct at these spatial frequencies. Indeed, recent cryo-ptychographic work, when optimized towards conditions for low-spatial frequencies, has demonstrated good information transfer across the low spatial frequencies present in amorphous carbon³². Coherent electron scattering from the amorphous carbon support, and any amorphous overlying contamination layer, may play a role in helping explain the high FRC scores we see at resolutions beyond what would be expected from information in the bright field disc, while operating in this so-called darkfield mode of electron ptychography used here. At the low beam energies we used, dynamical (multiple) scattering of the electron beam is more likely than at the

higher energies typically used in TEM. Thus, with an overlying aC-like layer, when the electron beam reaches buried crystalline regions, it no longer has the form of the electron beam in the vacuum: its angular form extends beyond that of the original brightfield disc of a direct, un-scattered beam. Similarly, an underlying aC-like support layer will multiply diffract beams scattered from the crystalline regions, potentially causing them to interfere. Thus, relative phase information between two otherwise separated crystalline diffracted beams could possibly be determined from the diffraction data using ptychography. Put another way, the support film and contamination layers diffract the illumination into a range of spatial frequencies which is much broader than the discrete peaks of the poly-crystalline gold regions, and we posit this may help explain the repeatability we see in the phase of the reconstructions of sample crystalline regions, at spatial frequencies beyond that expected from simple consideration of the brightfield disc. Without this mechanism, or other means to consistently determine relative the phases between otherwise separated diffraction discs, ambiguity in reconstructed phases may result.

This results for example in the fact that atomically sharp interfaces cannot be reconstructed with dark-field ptychography, as seen in the simulation in Fig 4. Some atomic lattice from the gold is leaking into the MoS₂ area, although we have perfect, simulated data.

Here, our model gold island did not have a sharp edge: it is tapered. This is described in the supplemental information, section 4.2 "... Au island subsystem which had a hexagonal frustum geometry". Also, it is evident if one looks carefully at Figure 4(d). Looking at the gold atom depictions (shown orange) on the edge we see 1 atom thickness, on the next row 2 atom thickness, then 3 atom thickness, etc. To clarify this in the Figure we have now modified the caption text to Figure 4:

d, Depiction of the simulated atomic positions at the edge of gold island, viewed from electron beam direction, where the sample is tilted at 2.9°. The gold island is modelled as having a frustum geometrical form (see Supplemental Information) and thus has a tapered edge, seen by the gradually increasing number of gold atoms at the edge.

Also in the Au/aC image, we do not see atomically sharp interfaces, and we can make out edges of gold particles where the lattice is leaking into the aC area.

Aside from our particles having a tapered edge, which may not have need noticed by the referee, we discovered an unfortunate error in our preparation of Figure 4(e). We are fortunate that this review process and the referee's comments gave us a chance to find and correct this. Our previous figure 4e had mistakenly been taken from an intermediate stage reconstruction we performed on single layer MoS₂ substrate with an Au Island, rather than our multilayer MoS₂ substrate. Related to this our registration between images was previously not accurate, and the ptychographic reconstruction had reduced correlation with the simulated plane wave illuminated exit wave (Figure 4c). We apologize for this error and hope that it does not unduly degrade the referee's confidence in our findings. The corrected figure 4(e) is now given. We believe that this does not show 'leakage' of the Au atoms beyond the expected boundary.

Please see the revised Figure 4 along with notes made previously regarding the particles having a tapered edge.

The following could be done to illustrate or refute this point: Instead of the FRC, which just compares two reconstructions without ground truth, the authors should plot the spatial-frequency resolved SNR(q) (eq 44 in <https://iopscience.iop.org/article/10.1088/1367-2630/14/6/063004>). For Fig. 4, ground truth data is available, so the SNR can be calculated. Also, the sample contains a sharp edge, so that all spatial frequencies should be present in the ground truth.

First, note that our sample does not contain a truly sharp edge. We have tapered ramp like-edges, and within these particles the potential is modelled as being produced from atomic potentials that are smooth and continuous. This is usually the case in multi-slice atomic resolution simulations – no sharp Heaviside or Dirac delta type functions are present. However, as the atomic potentials are significantly ‘sharp’ one would expect some intensity throughout the whole spatial frequency range of the image. Of more concern to us that our simulations currently do not actually represent our experimental situation, in that a structured amorphous carbon support film, and surface contamination, is not included. Including this support film and contamination layer in a complete simulation would (as mentioned above) make a good subject for a future and more detailed study.

For this paper though, inspired by comments from the referees we have now made a simple model of single slice test object which we show in a new Extended Data Figure 8. Here, we constructed this object to have a wide range of spatial frequencies, as would be present in a real sample covered by or supported on amorphous carbon. We use the same pixel size, illumination condition (i.e. same beam half angle, defocus, and wavelength) as the experimental data sets. Reconstructions were then made on the simulated data using the same recipe parameters as used on the experimental data, with the exception of this being adapted for a single-slice model.

Here we used a single-slice model, for simple illustrative purposes, to allow us to adopt exactly the same method as in the reference given by referee (<https://iopscience.iop.org/article/10.1088/1367-2630/14/6/063004>) which used a single slice model in order to allow an unambiguous and simple definition of SNR and PCTF. For the 3D dimensional case of multi-slice ptychographic models, consensus is still forming on how to most meaningfully and usefully define SNR and PCTF (usually presented as 1D characteristic) for a 3D / multi-slice case. Subtleties arise over whether comparison should be made between phase sums, amplitude products, complex products, or multi-slice propagated waves, or individual slices, for example.

However, going back to the request, we plot the calculated SNR for our simulation in Extended Data Figure 8f, which shows no gaps in the SNR characteristic, and SNR in the range $10^4 - 10^1$ out to approx. 1.5 \AA^{-1} . This adds some simulation-based credibility to the resolution we claim to have realized in our work. The PCTF in these simulations is also approximately unity out to approx. 1.5 \AA^{-1} , again consistent with our experiments.

Also in passing, we note that SNR and FRC are not entirely statistically independent. Optics Express, Vol. 26, Pages 3108-3123 (2018), Equation 34, and Methods in Enzymology, Volume 482, 2010, Pages 73-100, Equation 3.15 give $SSNR = 2 \text{ FRC} / (1 - \text{FRC})$. Thus, our FRC measure is expected to be linked the SNR. Ideally, of course we would have an exact model or ‘perfect image’ of our sample to directly determine SNR, but that in essence is part of the central problem of electron microscopy: all imaging methods have their own artefacts, and model methods have limitations too. A simple 1D - PCTF may not accurately represent the otherwise repeatable and reliable transformation that ptychographic imaging modalities produce. The point is whether we can get reliable and repeatable agreement between models and experiments (by some means which might be more complicated than a linear 1D-PCTF, which itself simplifies the physics of interactions) to allow us to usefully interpret or otherwise use the experimental data to gain useful information on the samples we observe.

Here we consider that our reconstructions meet this requirement of providing useful information at the resolution given, but in our initially submitted manuscript accept that we had may not made sufficient consideration of the mechanism for the repeatability, or explained that simple PCTF or SNR on 2D image

produced from the multi-slice reconstruction, may adequately represent the availability of the information. Regarding the information being produced more reliably than anticipated, we posit that the role of our support substrate or overlying contamination plays a role: we must consider the sample as a whole. We believe we have addressed this issue now, through the following modifications and additions to the manuscript which follow on from the additions on pages 8/9:

While a further study involving multi-slice simulations would be required to completely confirm this proposed mechanism, here we simulated the limiting case of a single-slice reconstruction of a simple mixed phase and amplitude object having a broad range of spatial frequencies and isolated resolution-testing double-dot like features (see Extended Data Figure 8(a, b)). The simulation used a beam convergence angle, defocus, illumination overlap, and reconstructed pixel size which were similar or identical to the experiment (see Extended Methods). The simulated reconstruction has a phase contrast transfer function (PCTF) characteristic (Extended Data Fig. 8e) with a steady near-unity value out to spatial frequencies $\sim 1.5 \text{ \AA}^{-1}$, and a SNR (Extended Data Fig. 8f), which diminishes from ~ 104 at low spatial frequencies to 10 at just over $\sim 1.5 \text{ \AA}^{-1}$. Noting the relationship between FRC and SNR^{49,50}, the $\frac{1}{2}$ bit FRC threshold would correspond to an SNR of 0.41 which appears at $\sim 2.2 \text{ \AA}^{-1}$ in these reconstructions.

Our experimental reconstructions, however, use a multi-slice approach, with a number of slices and separation that is towards the outer bounds of yielding a useful approximate multi-slice reconstruction⁵¹. The finite experimental electron dose, possible beam positional error, and the relatively few slices in our reconstruction likely contribute to our experimentally estimated repeatable information limit being less than from the single-slice simulation.

As comparison, perform a reconstruction of the same sample in Fig. 4 with the convergence angle chosen such that the first diffraction order disc and bright field disc have some overlap and compute the SNR(q). Can this "standard" ptychography setup reconstruct spatial frequencies which the dark-field approach cannot reconstruct?

In our new Extended Data Figure 8, and in the work surrounding that, we have considered a range of beam convergence angles, α . The larger beam convergence angles given would result in beam overlap if applied to the experimental samples. For our reconstruction recipes, we do see differences in the PCTF and SNR with α . When alpha is larger we see some improvement in SNR at low spatial frequencies but this diminishes at higher frequencies. Reasons behind this could be the subject of another paper: the behaviour appears non-trivial. We posit that there might be some influence from the phase of a high-alpha defocused probe varying rapidly towards its outer limits, and that this high-spatial frequency phase variation in our defocused probe might not be adequately sampled at higher spatial frequencies, resulting in potential inaccuracies.

Regarding the range of angles that have been considered see the new description in our extended methods (page 24):

Our simulated images and diffraction data for the Au – MoS₂ sample model were generated using Prismatic 2.0 software, using a multi-slice approach⁶⁶⁻⁶⁹ (see Supplemental Methods for more details). The single slice synthetic test object (phase shown in Extended Data Figure 8 a, b) was prepared using Adobe Illustrator software combined with Matlab to overlay a series of double-gaussian peaks with variable separations at well separated locations throughout

the image. Example double-peak features are at $x = 24$ nm, $y = 26$ nm in Extended Data Figure 8 a, b, c, d and shown with small bounding rectangle, which is shown in the expanded views of Extended Data Figure 8 h, i, and j. The object was computationally ‘illuminated’ with a beam of the same energy (wavelength) as in our experiments, and with convergence half-angles, α , defoci, and overlap that were approximately in the same range as our experimental conditions. Specifically, $\alpha = 7.5, 10, 15$ and 20 mrad were used in the simulations, where the latter two of these conditions would have resulted in overlap between the Bragg diffracted beams from their respective samples (see Extended Data Figure 8). These data were then reconstructed using the same parameters as for our experimental reconstruction of the Au-aC or Au-MoS₂ sample. The resulting reconstructed phase was then Fourier transformed and compared to the phase of our source object to determine a radially averaged signal to noise ratio (SNR), using the method in Eqn 44 of Thibault *et al.*⁷⁰, and phase contrast transfer function (PCTF) (Extended Data Figure 8 e, f).

I think this point is important and can be evaluated with existing code and infrastructure, thus it should be not too much extra work. Discussion of the results should improve the understanding of this optical configuration of ptychography.

>> We agree it has been worthwhile to investigate this point, though it took us a little longer than hoped to perform calculations to illustrate the point. However, as we noted previously a complete and accurate simulation of our system would include amorphous carbon, which would be a lot of extra work – and be another paper. We made some inroad on this discussion by talking about the role of carbon and support, and including a model of a synthetic test sample that contains a wide range of spatial frequencies. We hope that this helps take this discussion further, adds some extra support to our result and claim, and also importantly interests more readers in following up on and applying our work.

See the previously noted additional text added to the manuscript.

Another point is the accurate position determination. The reviewer thinks that uni-cell hops cannot be resolved in dark-field ptychography, which may also be another cause of lattice delocalization, like in Ref 33. If the authors think that they can reliably resolve unit-cell hops with their algorithm, they should show it with simulations. If not, they should point it out as future research to the reader.

Again, the referee raises an interesting point. We attempt to address this in our simulation presented in Extended Data Figure 8 h, i, j, k. Within our simulations we included some isolated separated point-like features (isolated double-gaussians). Though this does not fully address the question posed, it gives some indication that when there are many spatial features present in the object (and so the object has a rich spatial diversity), accurate localization of isolated features appears possible. In the case of highly repetitive (low diversity) crystalline structures accurate localization of such features by dark-field ptychography may indeed not be possible. Again, we make the point that experimentally the support film or contamination is likely playing a role in removing ambiguity in the reconstruction. We have pointed this out as area for further research.

Minor points:

1) since the setup is non-standard, a Ray diagram of the optics would help the reader understand configuration

We have now included a new Extended Data Figure 10, that outlines the optical configuration of the experiment.

This is reference in the main text, page 5:

In line with our goal of developing a high-resolution technique that could be applied to lower-cost general SEM/STEM class instruments, our projector lens system contained only a single lens (see Extended Data Figure 10) that was optimized towards size, cost and convenience, rather than for distortions – the primary one being pincushion – which we could correct digitally.

2) line 15 "Furthermore, lower electron beam energies are expected to give an increased information coefficient, provided the specimen is commensurately thinner and composed mainly of light elements."

The reader cannot be expected to know the paper that defines the term "information coefficient", please replace with standard, known terms.

In the abstract (opening paragraph) we have now used the phrase 'information per unit damage', and later in the text give the definition of information coefficient before continuing to use the term 'information coefficient' elsewhere in our text. See in the abstract / opening paragraph:

Furthermore, lower electron beam energies are expected to give increased information per unit damage, provided the specimen is commensurately thinner and composed mainly of light elements.

And on page 4:

Aside from the economic reasons for adopting ≤ 30 keV electron beams for sub-Å imaging, a recent study pointed out that the greater elastic scattering cross section (σ_e) achieved with a lower energy beam gives an improved image contrast which on balance can outweigh the effects of increased sample damage at lower energies²³. The information coefficient²³, defined as

$$\zeta = T \frac{\sigma_e}{\sigma_i}$$

where T is the total transmission through the sample and σ_i is the inelastic scattering cross section, is optimized by reducing the beam energy as the sample thickness is reduced.

3) line 27 "and in the longer-term assist in the structural determination of proteins with masses below about 100 kDa ..."

It should be mentioned that a different optical configuration with much larger convergence angle is necessary than

presented in this study, since proteins are weakly scattering and produce much less dark-field signal. The resolution will be limited to a little more than $2\lambda/\alpha$ (convergence angle)

It could also be argued that the 'probe' reaching a protein which is buried in an amorphous material (e.g. vitreous ice), is no longer just a simple probe with a circular aperture defined angular extent. We admit that current algorithms will have difficulty in making use of the weak darkfield signal, but it is not beyond the realms of possibility that the future fusion of single particle analysis techniques and multi-slice ptychography, could make use of these weak signals. To say that is completely impossible without further investigation of these possibilities, might stifle the required future studies to prove (or disprove) the referees assertion. Thus, we have modified the requested passages as follows:

(on page 14):

However, in the case of proteins, the dark-field signal arising from electrons scattered outside the brightfield disc is expected to be weak. At the low electron dose conditions required for non-destructively imaging proteins, advances in current ptychographic algorithms likely fused with single particle analysis methods would be required to make effective use of the weak dark field signal and to extend the resolution beyond the bright-field disk limited Abbe resolution of $\lambda/2\alpha$. Consequently, though our work in its current form cannot directly result in atomic resolution protein images, it demonstrates some the potential of 20 keV electron ptychography to yield sub-Å information that may assist in future structural determination methods.

In the length-restricted opening abstract paragraph (page 1 – 2) we have made a shorter modification to the reflect this as below:

“...of 2D material-based devices. In the longer-term with suitable beam conditions and improved reconstruction algorithms this may also assist in the structural determination of proteins with masses below about 100 kDa, which are abundant in nature but are difficult to characterize.”

4) line 76 "Reprocessing the data from these early experiments using an algorithm to account for the non-ideal camera response yielded some phase reconstructions which perhaps indicated the 022 planes (with 1.44 Å separation) of gold particles, though this was not confirmed using standard resolution assessment methods."

The feature delocalization could be mentioned here, as visible also in Fig. 1,3, and 4 of this paper.

We pointed out the tapered edge of our modelled particles in Figure 4, and also update figure 4 following a misplaced sub-figure as described earlier. Thus, in our interpretations and figure we consider that this does not show delocalization. As for Figure 1 and 3, which come from experimental data, here we have no definitive ground truth for the situation of species in the sample to specifically note features as delocalized. Any apparent 'delocalization' here does not have exactly the same form as might be expected from, say a non-aberration corrected CTEM, image. Thus, we would prefer to defer discussion of artefacts that may be reminiscent of delocalization to a future more detailed study (involving amorphous material interactions), rather than attempt to inadequately describe it here.

5) line 63 "Also, it was recently applied to electron microscopy of few layer TMDCs (e.g. MoS₂, MoSe₂, WS₂) and thicker samples of PrScO₃ [17-19] to give sub-Å information with a 80 keV[19] and 300 keV[17] beam energy, where it provided a 3 – 4x improvement over the conventional resolution of the electron microscopes used in the experiments."

Recent publications making use of the depth-sectioning capabilities of MS-ptycho should be included, e.g.

18 H. Sha, Y. Ma, G. Cao, J. Cui, W. Yang, Q. Li and R. Yu, Nature Communications, 2023, 14, 162.

19 H. Sha, J. Cui and R. Yu, Science Advances, 2022, 8, eabn2275.

20 H. Zhang, G. Li, J. Zhang, D. Zhang, Z. Chen, X. Liu, P. Guo, Y. Zhu, C. Chen, L. Liu et al., Science, 2023, 380, 633–638.

We had initially focussed our examples on TMDCs, but yes could have said more about other examples. We have thus added the references as requested, on page 3:

80 keV¹⁹ and 300 keV¹⁷ beam energy, where it provided a 3 – 4x improvement over the conventional lateral resolution of the electron microscopes used in the experiments. The depth resolution using multi-slice ptychography is also significantly improved over conventional STEM imaging, going well below 5 nm^{17,20}. This depth resolution capability has been advanced²¹ and used to reveal depth-dependent crystal structure orientations around dislocation cores²¹ and inhomogeneity in zeolites²², for example.

Here the new references are (as requested):

20 Sha, H., Cui, J. & Yu, R. Deep sub-angstrom resolution imaging by electron ptychography with misorientation correction. *Science Advances* **8**, eabn2275 (2022). <https://doi.org/10.1126/sciadv.abn2275>

21 Sha, H. *et al.* Sub-nanometer-scale mapping of crystal orientation and depth-dependent structure of dislocation cores in SrTiO₃. *Nat. Commun.* **14**, 162 (2023). <https://doi.org/10.1038/s41467-023-35877-7>

22 Zhang, H. *et al.* Three-dimensional inhomogeneity of zeolite structure and composition revealed by electron ptychography. *Science* **380**, 633-638 (2023). <https://doi.org/10.1126/science.adg3183>

We have also made additional use of these references on page 9, and page 11:

6) Extended Data Figure 7: why are regions excluded?

"To eliminate the contribution to our resolution measure of subregions that had little or no useful information (such as regions containing mainly amorphous carbon), regions with a mean FRC below a threshold given by Otsu's method were excluded from further consideration."

As noted in the main text, it is commonplace in the biological EM community to exclude the support material around the region of interest (usually a single particle) from analysis when determining FRC scores. See for example the reference we gave: 48 - Penczek, P. Reliable cryo-EM resolution estimation with modified Fourier shell correlation. *IUCr* **7**, 995-1008 (2020). <https://doi.org/10.1107/S2052252520011574>, where we quote:

"For the vast majority of practical applications it is necessary to restrict the real-space region of support to the area m occupied by the structure of a complex, and thus to exclude surrounding noise. m is a real-space function of the same size as the half-volume and, in the ideal case, is binary, i.e. contains only ones that indicate the structure and zeroes elsewhere."

This is largely because the correlation between amorphous carbon regions is weak and contains little information of use to the microscopist. However, that is not to say that it does not contain any information –

indeed by taking the FT one can determine the CTF of the system. Note that PCTF, SNR and FRC (which is related to SNR) all tell us different things about the system. See for example Extended Data Figure 8 e, f.

Other authors use amorphous carbon to get CTF in standard ptychography.

Yes, one can get the PCTF/CTF by analyzing the FT of the phase of a reconstruction of an amorphous material, or in principle any object that contains a very broad range of spatial frequencies. We used the latter approach of creating an artificial image with a broad range of spatial frequencies to prepare the data in the new extended data figure 8, e and determine a PCTF.

In passing we note that the application and determination of a PCTF assumes the sample is a simple phase object. Thus, computing the 1D radial average PCTF of a single image created by some means from a multi-slice ptychographic reconstruction of data created from a multi-slice model approximates a more complicated transfer function that should consider the full 3-dimensional nature of the sample and multiple scattering. Thus, though the PCTF given here (in our new extended data figure 8) is formed from using single slice model and the reconstruction is thus approximate, using a multi-slice method and representing its behaviour in a simple PCTF would also be approximate. In a sense, neither approach represents the physics of the situation fully; on other hand, both approaches (including our single slice model) offer a somewhat useful approximation.

The reviewer thinks this may be a result of the gaps in spatial frequency transfer. If this is so, it should be pointed out to the reader, see main concern.

As noted in the main text and above, it is commonplace to exclude the support material around the regions of interest (usually a single particles) from analysis when determining FRC scores. Compared to the information and signal contained from the crystalline material which concentrates intensity at nearly discrete spatial frequencies, the broad-spectrum amorphous carbon spectral information signal is weak. We believe there may be correlations in the amorphous carbon region, but we would need to use a much higher electron dose to see them. Using a much higher electron dose then poses another set of problems such as beam induced damage and/or contamination. We noted this in the additional text added to page 8 to address the reviewers initial major concern. Please refer back to our responses back on page 3 – 4 of this document, noting in particular the part we give again here:

The non-crystalline structure of the amorphous carbon, where the structural information is broadly spread over a wide spatial frequency range, would require a greater electron dose to reliably determine its structure and show correlation between independent scans in comparison to crystalline gold which concentrates most of its scattered intensity into or near discrete peaks (or Bragg discs) that are easier to distinguish in the presence of inevitable dose-limited shot noise. Furthermore, the high electron beam intensity in a CFEG STEM system can readily induce structural rearrangement and instability in aC, through mechanisms such as electron beam sputtering and deposition⁴⁸, which further reduces the likelihood of correlations between reconstructions of amorphous carbon.

7) "Defocused probe electron ptychography, as used here, does not require a high-resolution, finely focused, or spherically aberration corrected electron probe to achieve a high-resolution reconstruction. "

Again, please discern between dark-field and normal ptychography.

We have done so on page 9 and as below.

Defocused probe electron ptychography, where the dark-field diffraction data is also processed in the ptychographic reconstruction (as used here) does not require a high-resolution, finely focused electron probe, which is typically used in high-resolution analytical STEM and SEM. However, we note that defocussing the probe would also lower the resolution of x-ray spectra mapping, if it is performed simultaneously with the ptychographic data acquisition.

[Note the changes in the later part of the above are to address comments from another referee].

Also, we note that the term 'dark-field ptychography' is not as unambiguous as it was (if indeed it ever was). Note that in some x-ray work (see for example <https://doi.org/10.1364/OE.23.016429>) dark-field ptychography can mean completely obscuring the bright-field beam before it reaches the detector, so that only the dark-field is used. However, in the electron microscopy <https://doi.org/10.1038/ncomms1733>, dark-field ptychography can mean processing the dark-field **and** bright-field information. Similarly, as there are many variants and practitioners of ptychography around the world, we would rather avoid the phrase 'normal' to avoid unnecessary assumptions. We used 'conventional bright-field mode instead' to be more specific than just saying 'normal'.

8) it should be mentioned that the dark-field optical configuration as used here will not result in atomic resolution protein images

We have amended the text in the closing discussion, page 14 to say:

Consequently, though our work in its current form cannot directly result in atomic resolution protein images, it demonstrates some the potential of 20 keV electron ptychography to yield sub-Å information that may assist in future structural determination methods.

We still posit that scattering from the material in which the material is deposited, which has a wider range of spatial frequencies than in the particle or material of interest can play a role in allow phase relationships between seemingly separated brightfield and darkfield disks to be determined with less ambiguity than would be the case without this consideration. That said, electrons in the darkfield are weak, and thus the electron doses required to use the darkfield information may potentially be destructive to the observed protein. Research exploring the role of the support medium in ptychography may lead to new insights. Image aggregating and averaging methods, such as are used in current single particle analysis methods for protein structure determination, may also be able to overcome electron beam damage effects. This complete approach, that uses ptychography as just one part of a more complex reconstruction and model refinement process, may allow atomic resolution models of proteins to be reconstructed from this technique.

Thus, yes, we concede that our work does not result in a method of directly imaging proteins at atomic resolution, but it may eventually be part of a process or route to providing atomic resolution information on proteins. The above noted amendment describes this.

9) what is the expected depth resolution of dark-field MS-ptycho? has this been evaluated by you with simulations? if not, point it out as future research.

We have not evaluated the depth resolution of dark-field multi-slice ptychography within our simulations. Indeed, this is something we plan to look at more in future research, particularly in combination with tomography. We now point this out as future research as below.

When a focussed probe is used, with a convergence semi-angle sufficient to provide overlap between all diffracted beams, the depth resolution is improved, and has been shown to surpass that of conventional STEM^{17,21}, heading below ~ 5 nm. In our work we have not directly determined the depth resolution, as our goal was to produce TEM-like 2D images of the sample, rather than full 3-dimensional models. Determination of our depth resolution would require further analysis and modelling, which will be part of future work.

10) "We experimentally show that a resolution of at least 67 pm can be achieved using ptychography in a non aberration corrected scanning electron microscope, operating in a transmission mode with a 20 keV electron beam"

Does it provide continuous transfer of spatial frequencies like standard ptychography? See main concern.

In the limiting case of a thin object that can be adequately represented as a simple phase – amplitude object, our simulations show that a continuous range of spatial frequencies are indeed captured. In the case of three- dimensional crystalline samples, if the aim is to determine complete angular sampling of the scattering matrix, this would require a broader angular illumination that could be provided from low convergence angle electron beam. However, we note that in a multi-slice treatment an overlying layer of so-called amorphous contamination (which actually has long-range and short-range order) in-effect produces a coherent probe which has an angular range greater than the initially incident electron beam, which subsequently impacts on the underlying crystalline structured sample. Underlying layers will also provide information that would effectively help in determining the relative phases of Bragg discs, again through multiple scattering.

We believe we have addressed this concern in the previous comments and additions, where we also discussed that a larger electron dose, and a more stable amorphous sample, would have likely revealed correlations in the FRC across the complete range of spatial frequencies between the dominant Bragg peaks.

Reviewer #2 (Remarks to the Author):

This paper reports an important advance in low-energy transmission electron microscopy (TEM). With the advent of greatly improved electron lenses, which use multipole fields to correct for the aberrations inherent in conventional round magnetic lenses, the resolution of electron microscopes has quickly improved over the last 20 years. Given these enhanced lenses, a major thrust of aberration correction is to lower the electron voltage from the typical TEM acceleration of 200keV down to 60keV or lower. At lower energy, the electrons have longer wavelength, and can only provide intrinsically lower resolution: aberration correction can compensate for this. The rationale of low keV is to reduce damage events in the specimen (although some types of damage can be increased at lower keV). This strategy is sometime called 'soft microscopy' and promises to open up lots of new science, especially the understanding of structures consisting of only a few atomic layers.

However, the low-energy aberration-correction approach has met with further difficulties relating to decoherence due to thermal magnetic field noise in the electron optics. It also requires very complex and expensive instrumentation.

This paper reports on a different approach to very low energy transmission microscopy: the application of ptychography within a conventional scanning electron microscope (SEM), which is used with 20keV electrons (extremely low for transmission microscopy). Ptychography is very well established in X-ray microscopy. It utilises diffraction patterns that do not need to go through a lens. A proof of principle of ptychography in SEM was demonstrated by Humphry et. al. in 2012 but I am not aware of any further work on this technique since then.

The headline result is that the authors' microscope obtains a resolution of 67pm. Given the extreme low energy of the electrons involved, this is an extraordinarily impressive result, which could deliver all sorts of new science. As the authors also observe, an SEM is a fraction of the cost of a conventional aberration corrected machine and has a much smaller laboratory footprint.

However, there are some improvements in the paper which I would like to see implemented, and which I describe in detail below.

1) I want to start with what I think is its biggest weakness: the measurement of the resolution that it has purportedly been achieved. Leaving aside for now the diffractograms in Figure 1c, this headline result depends upon the line plot in Figure 3e, which itself depends on Figure 3c. Looking in detail at Figure 3c we can identify three distinct areas. Let us call the area with the oval marker, area 'A'. At the top of the figure, at about 15 degrees right from due North from the oval, there is a hexagonal crystalline area of a few unit cells, call it 'B'. At about 80 degrees from due North from the oval, and about halfway to the edge of the image, there is another crystalline area, call it 'C'. Area B appears to have two to three adjacent crystalline hexagonal unit cells of the same shape/size as the underlying MoS₂ at the bottom left of the simulation in Figure 4b. Area C is also crystalline, but with a much smaller unit cell, possibly also hexagonal: structures like this do not seem to be visible in Figure 4e, but are seen in Figure 4b (the untilted simulation). Meanwhile area A, which is used to confirm the real-space observation of the resolution record, is semi-amorphous, or at least very confused (with neither structures B or C prominent).

What is going on here? We know the two layers are incommensurate, giving rise to the Moire pattern in Figure 4a simulation, but the scale of the variation of the fringes is much larger, I think, than the field of view in Figs 3c/d, and so it seems unlikely that the Moire effect can transition so quickly from B to C. The only theoretical calculation of relevance is Fig 4e – the ptychographic reconstruction that should correspond to the experimental images figs 3c/d. It would help if we could be shown the intensity of Figure 4e, in order to make a direct comparison to Fig 3c, from which the actual line plot proving the resolution has been taken.

We thank the referee for their observations above. First, as noted in the comments and responses to referee number 1, we apologize for the error we had with Figure 4e. We have now corrected it. Also, we have now produced a new **Extended Data Figure 9**, which shows the amplitudes of the experimental and simulated reconstructions (thus combining elements of Figures 3 and Figure 4, in an amplitude presentation as suggested). In our opinion, and we hope that of referee now, this new figure (EDF 9) – with the amplitude presented as greyscale only – is less potentially confusing than the mixed phase-amplitude hue-brightness scale. Thus, the region we have drawn the line profile over appears less confused or ‘amorphous’ as the referee described it.

On a different tack, it is argued that the diffractogram in Fig 1c, and the corresponding ring correlation in Fig 2, proves the 67pm spacing is present in the image. It is true diffractogram analysis is often used to prove the resolution capability of a conventional TEM, but then the image is generated by an ‘analogue’ process: the image is formed directly by a lens. Here, the image is a reconstruction generated by applying a minimisation algorithm to very indirect data, which is never 100% reliable. There is therefore the possibility that the spacing is an artefact of the reconstruction. If this separation is indeed present in the real space image, the relevant fringes may not be properly localised at their real position (an effect that also happens in lens-based TEM – the presence of fringes of a given spacing is not the same as usable resolution at that spacing).

I think the authors should unpick these issues, because as it stands the structures visible in the relevant images are neither fully described or explained, and hence the resolution record claimed is not very secure.

We understand the point the referee is making here. Ptychography, as with all practical imaging methods, outputs images, or 2D/3D dataset, that have artefacts. With CTEM imaging there are well known artefacts, such as delocalisation that can occur due to defocus or spherical aberration, for example. Ptychography also has its artefacts, though the origin of the artefacts in ptychography are not always as physically simple to understand as is the case CTEM. Some ptychographic artefacts have been described and their origins previously illuminated (see for example https://doi.org/10.1007/978-3-030-00069-1_17). However, perhaps more could be done to try to classify artefact types root out their causes or correlations with factors in reconstruction parameters, or source data.

Another approach, that circumvents some fuller consideration of artefacts is to assume they are randomly distributed, such that only features that correlate between independent datasets and reconstructions should be trusted and used. Single particle analysis techniques that operate on thousands of images of the same system, in a way makes this assumption, and so FRC (or FSC) scores which look at repeatable information between images, models or reconstructions, have become a useful guide to the resolution of the information.

We note that the FRC based measure of resolution differs in definition from classical Rayleigh-like resolution measures that give a measure of resolution based on the ability of the system to delineate two discrete point-like features. However, in the presence of inevitable noise and finite sized particles, even ‘simple’ Rayleigh-like measures are not simple to reliably apply (<https://doi.org/10.1093/jmicro/dfaa018>).

The point of the discussion here is to highlight that resolution appears to mean different things to different people and in different contexts. We have tried to cover it from multiple perspectives (FRC, Rayleigh type line profiles, and image Fourier transforms), so that as a whole the overall picture of useful information or ‘resolution’ at the number we give becomes useful performance metric. Nonetheless we have also attempted to address the referees concerns by introduction of new figure (Extended Data Figure 9), and the references to it in the text.

As previously mentioned, the new Extended Data Figure 9, which shows the amplitudes of the experimental and simulated reconstructions. In particular, EDF 9d now shows line profiles from the experimental reconstruction, the ptychographic exit wave and the model plane wave exit. Again, we note that our model

has a tapered edge, as is expected for the experimental case. The agreement between the model and experiment is only approximate, as we have not attempted, for example, comprehensive and full 3d dimensional model optimization to get optimal agreement with our experimental data. However, the key features of there being there being alternate ~ 2 nm hexagonal regions that contain features separated by approx. 63 pm is present in all images.

Amendments and additions and deletions related to the above now span pages 11 to 12 and are given below.
First the deletion:

~~Looking at enlarged views of the reconstruction (Fig. 3c, d), features directly related to this 63 pm spacing can be directly identified. An intensity profile (Fig. 3e) from the region marked in Fig. 3d resolves two features, showing a mid-gap to peak intensity ratio of ~ 0.6 . Determining if this is a reliable resolution measure would ideally require having an estimate of the signal to noise (SNR) ratio in the region⁴⁷. However, here it is hard to estimate the SNR, so drawing the conclusion of 63 pm resolution from a single line trace or FFT would be subjective. Thus, for an objectively defined resolution measure that considers noise, we thus defer back to the FRC measures taken on Au/aC, which yielded 67 pm, but note that our MoS₂ reconstructions indicate this resolution could be improved upon.~~

[The original paragraph (above) has been recast to the one following the below]

We further make comparisons between the experimental reconstruction and a simulated reconstruction, created from diffraction data produced from a multi-slice model of the sample. Here 5 layers of MoS₂ and an angle of 2.9° between MoS₂ c-axis and the beam axis gave the best fit with the observed diffraction data, and the Au island was modelled with a tapered edge thickness (see Supplemental Methods, Section 4.1). If our experimental arrangement allowed us to realize an on-axis condition (see Methods), and plane wave illumination was provided, the exit wave in Fig. 4a, b would be expected. However, at our experimental sample orientation we are not looking directly down atomic columns (see schematic in Fig. 4d), and adjacent atomic columns begin to overlap with each another in a depth-projected image. The result is that the plane wave illuminated exit-wave (Fig. 4c, and Extended Data Fig. 9c) and simulated ptychographic reconstruction exit wave (Fig. 4e, and Extended Data Fig. 9b) from the tilted sample model do not allow the smaller hexagonal motif in the on-axis case (Fig. 4b) to be clearly seen. The exit-wave phase-shift determined from ptychography (Fig. 4e) differed, as expected, from a plane-wave illuminated TEM image exit-wave (Fig 4c). These differences were expected given the relatively low number of slices used in the reconstruction, as discussed in Methods.

Looking at enlarged views of the reconstruction (Fig. 3c, d, and Extended Data Fig. 9a), features directly related to an approximate 63 pm spacing can be directly identified. Similar features are also present in the simulated reconstructions (Extended Data Fig. 9b). Intensity profiles from experimental and model reconstructions (Fig. 3e, Extended Data Fig. 9d) appear to resolve two lattice-fringe features, showing a mid-gap to peak intensity ratio of ~ 0.6 . Although our models and reconstructions are in approximate agreement, the features present in the Moiré pattern

produced from the Au on the MoS₂ do not represent on-axis uniformly separated single atomic columns, given the tilt angle of the multiple layers in our sample (depicted in Figure 4d). Even in the case of an on-axis sample, where aligned atomic columns act as the point-like objects to be resolved, a reliable resolution measure should include consideration of the signal-to-noise (SNR) ratio in the region, which is not trivial to determine⁵⁷

Thus here the line profiles (Fig. 3e and Extended Data Fig. 9d) are included to add support the argument of there being useful repeatable information from independent reconstructions in the region of the maximum resolution (~67 pm) that is determined from the FRC method. The resolution measure from the FRC method, as it involves multiple reconstructions and consideration of noise, appears to us as the most useful and reliable resolution measure. However, we recognize there are many other resolution measures in use in electron microscopy^{49,58-60}.

Within the context of the FRC resolution measure, our work shows the capability of ptychography using so-called dark-field scattered electrons to yield information in the reconstructions that is beyond the Abbe resolution limit. As previously mentioned, simulations presented in Extended Data Figure 8 (h-k), further support this finding by showing that two isolated peak-like features, separated by around 67 pm can be resolved in a reconstruction, when using a beam convergence angle of ≤ 10 mrad with a 20 keV electron beam, as was used in our experiments

To further clarify our resolution claims, we have also adjusted the language so that we claim a ‘resolution measure’, rather than a distinct and singular ‘resolution’, as we note that there are multiple resolution measures in use in SEM, TEM, and electron tomography. See above:

... we recognize there are many other resolution measures in use in electron microscopy^{49,58-60}.

Following this, whereas before we talked about a ‘resolution’ now we say ‘resolution measure’. In the closing discussion we now say:

We experimentally show that a resolution measure of at least 67 pm can be achieved using ...

In the opening paragraph / abstract, we now say:

Here we demonstrate a resolution measure of 0.67 Å

We consider that the above additions and clarifications to the text address the referees concerns.

2) The details of the projector lens are not described. First, I think it would help the reader if the relationships between probe size, sampling in the diffraction pattern, total size of the diffraction pattern and resolution were outlined. What range of these parameters can be reached with the projector lens? (10 degrees is mentioned – does it have to be that big? How far is the projector from the objective? – I presume there must be a cross-over below it. I

personally would like to see a ray diagram, if only in the supplementary information. This is, after all, a key innovation in the experiment and the entire technique.

Given the maturity of ptychography now, we feel it is reasonable to point one of the many review type papers that cover the basics of ptychography, rather than go over basics here. In particular, we have referenced an open access publication [13, <https://doi.org/10.1093/jmicro/dfa055>] and https://doi.org/10.1007/978-3-030-00069-1_17. Some of the basics were reviewed in the supplemental methods, such as the relationship between angular field of view on the detector and reconstructed pixel size. See the bolded text below appearing on page 5.

Here, to achieve sub-Å imaging we aimed for a ~25 pm reconstructed pixel size. To obtain this reconstructed pixel size requires collecting electrons scattered by up to ~10 degrees (~175 mrad) when using a 20 keV beam (see **Supplemental Methods, Equation S.2**)¹⁵.

where in the supplementary methods, we stated:

... The reconstructed pixel size d is

$$d = \frac{\lambda}{\Omega}, \quad (\text{S.2})$$

where λ is the electron wavelength, and Ω is the angular field of view on pixelated detector at the diffraction plane¹⁵

We have also now created and included an additional Extended Data Figure 10, which helps explain the arrangement of our microscope. Full technical details cannot be disclosed due to this being a commercial prototype and experimental instruments, though the figure outlines its key features. In particular answering on the of the referees questions, yes, we form an additional cross-over below our projector lens. In our lens arrangement, a key aspect is that the maximum scattering angle that can be recorded is limited by our fixed position HAADF detector, as can be seen in Extended Data Figure 10.

Depending on the exact sample positioning, this limits the scattering collection angle to about 15 – 20 degrees. The angle of 10 degrees from the axis we gave corresponds to 174 mrad, or 350 mrad angular FOV, giving about 25 pm reconstructed pixel size at 20 kV. Our reconstructed pixel size was approximately 24 pm. Ultimately electrons must be scattered from the sample at the range of this angle for us to achieve an adequate reconstructed pixel size to reliably claim a resolution down to around 50 pm with adequate sampling. Thus, yes, our scattering angle needs to be this big for demonstration of the resolution presented here.

Projector lenses can magnify the diffraction plane, and effectively reduce the angular pencil angles of the rays making up their image. Thus, the aberrations present in the objective lens field region dominate. If there are distortions at this back focal plane, they are generally not at all straightforward to remove, as discussed later.

The new figure is referenced in the main text as below:

Page 5 – (Main Text)

In line with our goal of developing a high-resolution technique that could be applied to lower-cost general SEM/STEM class instruments, our projector lens system contained only a single lens (see Extended Data

Figure 10) that was optimized towards size, cost and convenience, rather than for distortions – the primary one being pincushion – which we could correct digitally. A well designed, higher-voltage dedicated STEM, would likely have less distortion than was the case with our experimental instrument.

Page 20 – (Methods)

Samples are placed in the immersion-objective magnetic-lens gap (see Extended Data Figure 10) using a conventional side-entry style transmission electron microscope (TEM) holder.

...

The instrument was modified from a standard design to include a small projector lens (item 6 in Extended Data Figure 10a) beneath the objective lens.

3) Does the large probe size help with reducing beam damage? The total flux/area through specimen will remain constant, but it has been proposed that scanning randomly could reduce time-dependent damage effects, or even local heating effects.

We do have not evidence of the large probe size reducing beam damage. Yes, proposals have been made of random scanning possibly reducing time-dependent damage effects, or even local heating effects. Reports effects seem subtle and relatively weak in the literature in our opinion, and we choose not to comment on these possibilities in our manuscript.

4) Why was the circular aperture constraint used during the reconstruction needed for one sample but not the other?

This was touched upon in the manuscript: the lower diversity in the MoS₂ sample diffraction data was prone to producing structure in the probe that mimicked the sample without this constraint. Specifically, the probe would acquire non-physical hexagonal motifs, which in turn resulted in non-physical three-fold and six-fold symmetric features in the diffraction data. Use of the aperture, equivalent to a physical aperture constraint, prevented these artefact from occurring. As the Au particles on aC data was much more diverse (specifically no repeating hexagonal motif throughout the dataset) this effect was much less prone to occurring. Thus, there was no need to include this constraint in the Au aC reconstruction. We have made an addition to the text to point this out:

To form a convergent and physical ptychographic reconstruction (Fig. 3a, b), we found it necessary to force pixels outside a certain radius in the FT of the electron probe model towards zero to gain a stable and physical probe and object reconstruction (see Methods). This constraint on the electron probe model, which was not required for the Au/aC reconstruction, corresponds to the beam being defined physically by a circular aperture. The reduced data diversity in the MoS₂ based diffraction data accounts for the need to apply this additional constraint; without it the reconstruction algorithm tends to assign a non-physical object-related periodic structure to the probe.

5) Shifting the probe can induce position-specific aberrations (coma, etc). Was there any evidence of this?

Over our relatively small fields of view, we saw no direct evidence of position-specific aberrations. Comparing slightly under- and over-focus HAADF images there appears to be uniform blur across the field of view. If there was a position dependent coma, the blur would vary across the field of view (assuming a planar sample which is perpendicular to the beam direction). However, we saw no evidence of this. Over larger fields of view than used in our experiments, say many microns (such as when the microscope is operated in a 'low mag' mode of operation), we would anticipate that position specific aberrations would be evident and would need to be accounted for in the probe reconstruction algorithm.

6) "We found that despite the diffraction data being distorted, a ptychographic reconstruction could be made, and its fast Fourier transform (FFT) yielded well-defined low-order diffraction peaks concentrated in rings, as expected for a polycrystalline sample." I note that in the case of putting a detector very close to the object, so that it is in the Fresnel condition, the ptychographic reconstruction comes out near-perfectly, even if a Fourier propagator is assumed in the algorithm. The difference is that the probe reconstruction develops a phase curvature reminiscent of a lens. This is because a pre-factor of a parabolic phase can be used to form the Fresnel integral, while still using the Fourier transform. The extra lens phase brings the parallel beams of the FT to a focus on the nearby detector. I say this because it is possible, I think, that the pin-cushion distortion may also be accounted for by a distorted reconstruction probe phase. If that's the case, the authors could avoid all the complexity of their pin-cushion mapping algorithm. No alteration to the paper required, just an idea for further work...

It is our understanding (supported by discussions with others knowledgeable in the field of electron optics) that pincushion distortions and other off-axial distortions or aberration of this nature cannot be accounted for through additional curvature added to the probe phase. One simple explanation is to consider that any phase curvature or modification to the probe or illumination in general is, through a Fourier transform, related to a convolution with a point spread function (PSF) that is applied over the image at the image plane. In the case of pincushion distortion (or other off-axial distortions, where the distortion varies with distance from the axis), the effective PSF would have to vary with position over the image [see for example <https://doi.org/10.1364/OL.451403>]. A positioning varying PSF cannot be represented by singular probe. This is also the issue underlying the previous comment, #5.

Thus, such distortions in the diffraction data cannot be absorbed in the probe function in this way. These are distortion are most easily handled using a image mapping approach.

We have added sentences to our manuscript to make this distinction clear and obvious. It is one of the core points of our work, so if the referee did not find this obvious we clearly need to address it. We have thus added the underlined sentences below in page 5:

"

... Thus, at these relatively large scattering angles we was expected to lead to so-called pincushion distortion in the diffraction data. This distortion is a primary aberration of round magnetic lenses, having a form where the ratio of the measured to actual distance from the centre of the diffraction plane varies in proportion to the square of the distance from the optical axis³⁸ (see Methods, Equation 1). This relationship is similar in form and related to spherical aberration which operates on the image plane^{38,39}. However, unlike spherical aberration in the electron-probe forming optics, which remains approximately constant over typical high magnification fields of view in the SEM, pincushion

distortion over the diffraction plane cannot be represented by a singular point spread function that is applied to the diffraction data. Thus, such distortions on the diffraction plane cannot be absorbed into the probe reconstruction.

Measurement and subsequent correction of the diffraction pattern distortion is thus essential to give meaningful and accurate reconstructions at the greatest resolution.

”

7) Somewhere in the paper I think the disadvantages of using a large probe should be mentioned, i.e. the loss of the analytical X-ray signal. Elemental determination is one of the biggest applications of SEM.

Given the length requirement for this article, full exploration of ‘pros and cons’ of the general defocused probe ptychography had not been covered, with it being assumed that fair consideration had been given to this in other reviews on the subject. But it is reasonable to point such things out. We have thus added the below to address this in our revised manuscript.

Defocused probe electron ptychography, where the dark-field diffraction data is also processed in the ptychographic reconstruction (as used here) does not require a high-resolution, finely focused electron probe, which is typically used in high-resolution analytical STEM and SEM. However, we note that defocussing the probe would also lower the resolution of x-ray spectra mapping, if it is performed simultaneously with the ptychographic data acquisition.

8) Caption fig 4d: There must be a better of saying ‘Depiction of the simulated atomic positions simulated, ...’

Thank you for bringing this to our attention. We have modified this to the simpler phrasing below:

“Visualization of the atoms at the edge of the modelled gold island.

9) Figure 1 has no 1d, although ‘d’ is in the figure caption.

Thank you for spotting the typo. It arose from an earlier draft of the manuscript. We have removed it.

There is no doubt that this paper makes an important contribution to the state-of-the-art of electron microscopy. It is technically sound and well written. I would strongly support its publication in Nature Communications. My only concern is the question of having proved conclusively the resolution that they claim has really been reached. The resolution is undoubtedly greatly increased over previous work, but is it by as much as the authors claim? More careful interpretation and discussion of the results is required, I think.

We thank you for your comments and recommendations. Also following from referees’ 1 comments, we now have a much deeper discussion of our results and possible mechanisms of what we are seeing here. We hope this satisfies the referees concerns.

Reviewer #3 (Remarks to the Author):

This is an interesting paper that reports Ptychography at low voltage. The paper describes several applications which demonstrate enhanced resolution at 20kV compared to other imaging methods and provides some details of the instrumental and computational requirements used to achieve this.

As a technical improvement this is worth publishing in a technical journal. However, in my view it does not represent the substantial improvement that is needed for the wider readership of Nature Communications.

I also have a number of specific comments / criticisms of the current paper.

1. Most importantly the work reported is very similar to the recently reported work of Allen et al. (APL, 2023) who showed that ptychographic reconstructions beyond a 2 alpha limit could be achieved for MoS₂ samples at 30kV. The authors reference older work by Rodenburg et al using an SEM but do not consider this more recent work which provides a more general and efficient method for ensuring convergence than that described on page 8.

The work by Allen referred to appeared in press only 2 weeks before we initially submitted it into the Nature publishing system. We apologize for missing it within our final round of pre-submission checks. We have now added a reference to this work. We note however that our work is at a significantly lower beam energy of 20 kV (0.667 of the work of Allen et al.) and is performed in a non-aberration corrected instrument. These are significant differences, and we believe that the manuscript still stands in importance owing to these major differences. That our work is performed in a non-aberration corrected instrument has all the advantages we have listed in our manuscript: reduced cost, ease of use, reduced footprint, and so presents a much stronger advance to the field.

Regarding the referees comment the APL 2023 paper provides a “more general and efficient method for ensuring convergence”, whether there is a strict definition of ‘efficiency’ here is not obvious. Given that our work uses and is aimed at lower cost instruments one could argue that our method is more efficient in terms of dollars per pm or nm. Anyway, the point of paper is not strictly ‘efficiency’, but rather just that it can be done and presents a new method (using distortion correction and low cost optics) of doing so. Both our work and the APL 2023 work use a super-resolution method, both use multiple probe modes, both use ePIE. The authors of the referenced APL 2023 do not specify or describe the parameters of their use of the ePIE algorithm (such as the alpha and beta ‘feedback parameters’, whereas we do specify them, so we cannot comment of details of differences.

However, our algorithm uses ePIE for only the first 50 iteration, then a maximum likelihood solver (referred to as MLs) for the remaining iterations as publications [M. Odstrčil, A. Menzel and M. Guizar-Sicairos, Optics Express 2018, 26(3), 3108] have shown MLs in general offer a better ultimate resolution limit compared to ePIE. This may in part explain our results where we argue that we show a superior resolution. Also note that our work used a different detector to that used in the APL 2023 paper, which comprised 4 times as many pixels, and it was tailored specifically to operate at low kV (down to 7 kV), unlike the Medipix3 detector. Anyway, we have added to the text:

...of gold particles, though this was not confirmed using standard resolution assessment methods²⁵. More recent work using a 30 keV electron beam, in a 3rd order aberration corrected STEM as opposed to a non-aberration corrected SEM, formed a ptychographic reconstruction from padded diffraction data demonstrated a resolution of at least 1.2 Å at 30 keV²⁸, thus providing a factor of 2 improvement on the first SEM-based demonstrations²⁵.

The new reference 26 is:

26 Allen, C. S., Danaie, M., Warner, J. H., Batey, D. J. & Kirkland, A. I. Super-resolution electron ptychography of low dimensional materials at 30 keV: Beyond the detector limit. *Applied Physics Letters* **123** (2023).
<https://doi.org/10.1063/5.0143684>

2. Throughout the paper there are unfortunate comparisons between the work reported and alternative imaging methods without reference to the differences between these. For example the resolution in an SEM is quoted as 5Å but this is for SE imaging which is not comparable to the ptychographic phase reported here.

We had perhaps been assuming a bit too much on the typical reader's understanding here. Clearly, yes, the imaging 'contrast' mechanism in ptychography is different from in SE imaging, the most common modality of SEM imaging. We have thus clarified the above by stating this resolution is for SE imaging. We have also added references [9, 10] so that readers can look more into what is meant by the resolution figures we gave. In line with the new references, we updated our 'about 5 Å' to instead say 4 Å as given in the new reference 9. Also, we are now more specific in saying that ptychography improves the resolution of TEM, rather than just electron microscopy.

Page 2

Consequently, low energy SEMs have become a ubiquitous and essential tool for micro- and nano- sciences, engineering, and technology. However, the resolution of a 30 keV non-aberration corrected SEM/STEM, is generally at most around 4 Å for secondary electron imaging⁹, and for STEM mode brightfield imaging. Si lattice fringes of 0.157 nm have also been observed¹⁰. While some aberration correctors and monochromators for ≤ 30 keV TEM^{1,11} and SEM^{12,13} have been developed in recent years to approach sub-Å resolution, they still retain the problems of cost and complexity. A relatively new approach to improving the resolution in TEM electron microscopy which does not require an aberration corrector or a high-pixel count electron detector is ptychography¹⁴.

Page 3

To make things even clearer now we have also stressed that the images produced by ptychography are not directly like SE images collected in a SEM, as below:

... data. The images it produces reveal information most similar to that produced from BF-STEM or TEM, though with addition of quantitative phase-shift information. It is not like secondary electron (SE) imaging in the SEM: it does not reveal surface topography or give SE yield contrast.

3. A large part of the paper is devoted to methods for measuring and correcting the diffraction patterns used. This is simply a feature of the relatively simple post specimen optics employed here and would not be a problem for more conventional multi lens projective systems which easily attain distortion free data at 99.9%. This discussion might however be useful in the context of a paper describing a low cost instrument for ptychography.

The major context of this manuscript is that our system presents a low-cost instrument for ptychography. For examples, see the 4th sentence of our abstract / opening paragraph. We highlight in bold sections where this was explained.

"In comparison, a low-energy (≤ 30 keV) scanning electron microscope (SEM) is simpler to operate and has a **reduced** footprint and **cost**."

and also see our original closing discussion:

“In the longer term, this may assist in the structural determination of proteins with masses below about 100 kDa, by **circumventing the potentially prohibitive financial cost**, space and personnel requirement of conventional higher beam-energy TEMs²,”

Further examples are on Page 2, Paragraph 3:

“However, an aberration corrected (AC-) TEM with an advanced high-pixel count direct electron detector has a financial cost, space, and personnel requirement that is prohibitive for many laboratories². In comparison, low energy (≤ 30 keV), non-aberration correction scanning electron microscopes (SEM) are smaller, **have significantly lower cost and are more economical to run.**”

Page 4, Paragraph 1:

“**Aside from the economic reasons for adopting ≤ 30 keV electron beams** for sub-Å imaging, a recent....”

However, we could have made it clearer that CTEM instruments employing more projector lenses might not need this distortion correction. However, this is not always the case particularly at the large (10 degree) scattering angles of this work. In routine or ‘casual’ electron diffraction work observationally it appears that diffraction pattern distortions are often overlooked, as often one is just looking for, say, a signature of a certain orientation or structure type, which can often be obtained from looking at the first 2 diffraction orders. However, in the field of structure determination by electron crystallography field it is well known that distortion corrections should be made to electron diffraction data for accuracy [<https://doi.org/10.1107/S2052252522007904>].

Finally we should point out that STEM performed in an image aberration corrected TEM (which one might naively expect to have low distortion) is often one of the worst offenders for diffraction plane distortion, as was pointed out by Jones [<https://doi.org/10.1088/1757-899X/109/1/012008>]. In this particular reference notice that the Kikuchi bands which should be straight are curved and have spiral distortion – which must physically be accompanied by pincushion distortion. Thus, it’s not always the case that advanced aberration corrected S/TEMs (or even conventional STEMs) with multiple projector lens system have low diffraction plane distortion. We have tried to address in the modification on page 5, given below:

Here, to achieve sub-Å imaging we aimed for a ~ 25 pm reconstructed pixel size, which thus requires collecting electrons scattered by up to ~ 10 degrees (~ 175 mrad) when using a 20 keV beam (see Supplemental Methods, Equation S.2)¹³. ~~As we added only a single magnetic round lens to our SEM to project and control the magnification of the diffraction plane on to our camera, this relatively large scattering angle was expected to lead to so-called pincushion distortion in the diffraction data.~~ In line with our goal of developing a high-resolution technique that could be applied to lower-cost general SEM/STEM class instruments, our projector lens system contained only a single lens (see Extended Data Figure 10) that was optimized towards size, cost and convenience, rather than for distortions – the primary one being pincushion – which we could correct digitally. A well designed, higher-voltage dedicated STEM, would likely have less pincushion distortion than was the case with our experimental instrument.

This Pincushion distortion is a primary aberration of round magnetic lenses, having a form...

4. There are many examples where comparisons / descriptions are only qualitative e.g. on p7 the reconstruction is

described as stronger and sharper with no values given. Similarly on p3 the statement that “ptychography requires the illumination to have significant coherence...” could easily be quantified.

The only instance we can find of the phrase ‘stronger and sharper’ is in the sentence:

“A reconstruction created using the undistorted diffraction data had a reduced Fourier error metric and the FFT of the reconstruction contained stronger and sharper peaks.”

In this sentence the quantitative part is the reduced Fourier error metric; a qualitative effect that generally accompanies this metric reduction is that peaks in the FT become more intense and sharper. In our opinion it would require quite a detailed study to do some correlated study of reduction in error metric with the quantitative measure of the intensity and sharpness of peaks in the FT. We focussed our efforts on quantifying the things that really mattered to this study, such as the resolution measures. We feel we had to draw the line somewhere and describe some things qualitatively.

Regarding coherence, as far as we can see in the literature there is no precise consensus on the quantitative level of coherence (defined in the longitudinal, lateral, and temporal senses) that is required for ptychography to be ‘successful’. Even expert reviews in the field electron ptychography (https://doi.org/10.1007/978-3-030-00069-1_17) don’t place precise values on coherence requirements and tend to use qualifiers such as ‘significant’ and ‘partial’ coherence being required.

From a practical sense in electron microscopy (as opposed to x-ray microscopy for example), it might actually be more useful to link the coherence requirements back to the electron source brightness, energy spread beam aperture arrangements and stability requirements, which ultimately bring about beam coherence. Generally, thus, CFE will be better than Schottky sources, which in turn will be better than electron microscopes with thermionic electron sources. We would argue that it is not yet precisely quantified just what the coherence requirements and consequences are in electron ptychography, particularly with the advent of multiple probe-mode ptychography. It would make an interesting subject for a future detailed study.

Yes, there are other instances where we have used the word stronger, but as above we feel we had to try to concentrate our efforts on quantifying the ‘end result’ (i.e. resolution measures), rather than all the intermediate qualitative indicators seen in intermediate steps and images.

5. The description of a lack of FRC data (p7) between the Bragg reflections is confusing – amorphous carbon does have structure and the FRC approach should have a positive value as the sample is the same for the two subsets of data correlated.

Here we refer the referee to the related points raised by referee #1, particular their point #6. We restate that here:

“Compared to the information and signal contained from the crystalline material which concentrates intensity at nearly discrete spatial frequencies, the broad-spectrum amorphous carbon spectral information signal is weak. We believe there may be correlations in the amorphous carbon region, but we would need to use a much higher electron dose to see them. Using a much higher electron dose then poses another set of problems such as beam induced damage and/or contamination. We note this in the following new text in the manuscript on page 7:

In Fig. 2 we additionally plot the average radial profile of the FT of the reconstructed intensity and the diffraction intensity profile expected from randomly oriented gold particles, where we label the dominant Au diffraction peak

indices. The non-crystalline structure of the amorphous carbon, where the structural information is broadly spread over a wide spatial frequency range, would require a greater electron dose to reliably determine its structure and show correlation between independent scans in comparison to crystalline gold which concentrates most of its scattered intensity into or near discrete peaks (or Bragg discs) that are easier to distinguish in the presence of inevitable dose-limited shot noise. Furthermore, the high electron beam intensity in a CFEG STEM system can readily induce structural rearrangement and instability in aC, through mechanisms such as electron beam sputtering and deposition⁴⁸, which further reduces the likelihood of correlations between reconstructions of amorphous carbon.

6. There are a number of clear typographical errors; more than I would have expected. For example p8 incorrectly describes inelastic and not elastic propagation and p9 quotes a 68.3 Å spacing.

We are sorry for this and have done our best to identify and correct any other typos appearing since our initial submission. Regarding the specific examples given above we have corrected them:

Regarding the probe reconstruction, we note that ptychography in its simplest and first developed forms assumes a single coherent illumination mode propagates ~~in~~elastically through the sample¹⁴.

From a reconstruction of the sample exit-wave and its FT (Fig. 3a, b) we see features with a spatial frequency of at least 1.46 \AA^{-1} , encircled in yellow, corresponding to the ~~68.~~0.683 Å spacing of the {401} plane group in the MoS₂.

Throughout the term FFT is used to describe the power spectra (or spectral densities) shown. This is incorrect the FFT is a computational approach (due to Cooley and Stucky) for calculating the discrete Fourier Transform of an object.

We used FFT following the perhaps unfortunate ‘common vernacular’ that appears popular in the electron microscopy community. We would speculate that this arose through widespread use ImageJ or Gatan Microscopy Suite (Digital Micrograph) software packages, whereby if one wants to find the power spectra or spectral densities of an image or dataset, then one clicks the ‘FFT’ button.

More correctly we should have (mostly) used the term Fourier transform, as determining the power spectral density of complex data would imply that we square the magnitude of resulting FT to give an equivalent to a real power or intensity. In most instances we were not determining a real power or intensity, which is why we went into some detail to describe and decide upon the which aspect of the FT to present in Extended Data Figure 6. Generally we presented the amplitude of the (complex) FT, not the amplitude squared (which would be the PSD). After determining the FT of images we often applied gamma corrections before presenting results, which are also detailed in the text. If we were to rename our FTs to PSDs we would thus need consequently relabel some of gamma factor labels by a factor of 2, owing to the definition of PSD, which would be a further and unnecessary complication.

As a compromise we have changed FFT to FT throughout the text and made a note that when we use the term ‘FT’ in our text we mean that we calculated the FT by using the FFT method. See the modified sentence from page 6 below:

We found that despite the diffraction data being distorted, a ptychographic reconstruction could be made, and its **Fast** Fourier transform (FT), which we found by using the Fast FT algorithm, yielded well-defined low-order diffraction peaks concentrated in rings, as expected for a polycrystalline sample.

Figure 1 a claims to show the exit wave but the data shown is real (I assume it is the phase) not complex as expected for the exit wave. The same applies to Figure 3 a.

Reading the figure caption for Figure 1:

“The color scale representing **phase** and **amplitude** inset within (b) applies also to (a)”,

and looking at the 2-dimensional hue-value color scale inset in (b), we thought it was reasonably clear that the data presented is complex. This presentation of complex data appears quite common in the literature. However, we have now tried to make it even clearer by saying

“The hue - value color scale representing phase (hue) and amplitude (value) of the complex data inset within (b) applies also to (a)”.

Similarly, we have modified the caption to Figure 3a, where we also noticed an unfortunate typo regarding a reference to color-saturation scale rather than hue-value scale:

The phase and amplitude of the complex quantities in a and d are presented using a hue - value color-saturation scale which is inset in **d**. where Here ϕ represents the phase in radians ...

Finally extended data Figure 8 does not directly add scientific value to the work presented and should be deleted.

We have removed this figure, though still believe that it did add some useful context to the work. Effective scientific communication we believe does involve significant work in placing the work in a broader context so that tax-payers etc who fund the work have a better chance in understanding its value, particularly in the current age of open-access, high-visibility publications such as nature communications. That said it is, of course, also important to get this work published for effective communication, so we follow the referee’s request.

If the journal editor is reading this far into our response and has any view on its value and whether to include this figure or not we would be happy to hear from them.

7. P10 contains a description of the sample mistilt which could have been corrected making the comparisons between simulations and experiment more meaningful. I accept that only a single tilt holder was available but given that the instrument used has a conventional side entry goniometer, surely using a commercially available double tilt holder would have been possible.

Despite our best efforts of fund raising and asking partners and collaborators it was not possible to gain a double tilt holder for this instrument. Unfortunately, the sample holder geometry for the instrument used in our experiments (the SU9000) is different to other Hitachi TEMs such as the other Hitachi TEMs at our facility. The holders are not cross compatible. So, unfortunately, we did not have funding available for this, and it was not possible.

When we talk about the need and desire to lower the cost of electron microscopy instrumentation to increase accessibility of the techniques in our manuscript, we personally feel and know this motivation. We are not talking about it in passing.

8. On p10 there is match between experiment and simulations that is used to obtain a sample thickness to high precision. However, the Stobbs factor is not considered.

The largest and most dominant physical factor that has relatively recently been shown to account for the Stobbs factor which relates simulation to experiment is the modulation transfer function (MTF) of the camera (<https://doi.org/10.1103/PhysRevLett.102.220801>). Once account is made of the camera MTF, good agreement can be made between simulated and experimental HRTEM images. In large-pixel hybrid-type direct electron detectors, such as the detector used in our work which has a pixel size of 75 μm , and such as the Medipix the MTF and Detective Quantum Efficiency (DQE) at low kV is close to ideally flat at low spatial frequencies (<https://doi.org/10.1016/j.ultramic.2017.06.010>).

Also note that comparing integrated intensities of the relatively large diffraction discs, such as we used to determine the sample thickness, relates to comparing features which are both at the low-spatial frequency end of the MTF. The Stobbs factor correction, being MTF related, is greater when comparing high-frequency signals with low-frequency ones, such as say when comparing lattice feature intensities in an image with the low spatial frequency 'background' signals. Thus, here even if our MTF is non ideal, we would not expect significant intensity correction to be required when looking at intensity ratios of similar low-spatial frequency signals.

We have added a note about the above reasoning in our supplemental material, which prior to this had just been implied. Please see the supplementary methods, section 4.1, for the new paragraph copied below:

As in the main text, we found best correlation with our models when 5 layers were used in the model. Comparing ratios of the integrated intensities of diffraction discs relates to low normalized spatial frequencies (<0.1) in the modulation transfer function (MTF) of the detector, owing to the relatively large size (~20 pixels) of the diffraction discs. Though we did not characterize the MTF of our pixelated direct detector at 20 keV, we suspect and assumed it would be similar to the near ideal characteristics displayed by Medpix detector at 60 keV⁸², owing to detectors having similar physical arrangements, albeit with our detector having a larger (75 μm versus 55 μm) pixel size⁵⁹. Owing to the expected flatness in this low frequency-end of the MTF, we did not consider it necessary to apply any Stobb's-like correction factor to in order match our simulations and experiments⁸³.

Also, though we had referenced section 4. 1 of the supplementary material in the main text, we had not done so in the Methods section. This will also help readers better access the above description.

However, here we determined that the MoS₂ was composed of 5 mono-layers which implies a thickness of 3.1 nm, using model-fitting between simulated and experimental convergent beam electron diffraction patterns, obtained by 4D-STEM (see Supplemental Methods, Section 4.1).

9. The manuscript contains an excessive use of superlatives – “new records” , “dramatic” etc. These are not needed.

>> We have reduced the use superlatives in the manuscript. See our deletions:

In the abstract / opening paragraph -

The ~~dramatic~~ drop in energy and resolution improvement is enabled ...

Page 14 -

This ratio appears to ~~be a new record in electron microscopy, exceeding recently reported values of this ratio⁶⁰ (see Extended Data Fig. 8) go beyond that of earlier sub-Å ptychography demonstrations⁶⁰, albeit where the information we obtain here is within a less constrained multi-slice approximation, where potential phase reconstruction ambiguities may occur when using thicker samples.~~ We achieve this ~~and also improves upon that of earlier sub-Å ptychography demonstrations~~ while being at ≤ 0.25 of the beam energy of these earlier sub-Ångström studies.

Overall if the above were addressed this work could be published in a specialist journal with, perhaps more emphasis on the instrumental modifications.

We will keep that in mind, if this submission does not work out. Thank you for your time and comments which have been valuable to improving our manuscript. We hope our additions to this work are sufficient to make you reconsider.

We also apologize to all referees for the time it has taken us to get these comments back to you and thank you all in advance for your further consideration.

Reviewer #1 (Remarks to the Author):

In Revision 1 the manuscript has improved significantly and many comments were resolved.

I acknowledge that the matter of information transfer in conjunction with multislice reconstruction requires more work to fully unravel, and the authors have made some strides and the necessary comments to point out the intricacies.

There are some remaining issues which should be resolved before publication - some slipped through the first round of the manuscript, some newly introduced in Rev1.

1) FRC depends on initialization - which initialization was used for the algorithm? If the same initialization was used for both independent reconstructions, this can lead to a biased FRC, as shown in [Allars, Frederick, et al. "Efficient large field of view electron phase imaging using near-field electron ptychography with a diffuser." Ultramicroscopy 231 (2021): 113257.]

>> Thank you for bringing this to our attention. We had been using a flat phase object for our initialization in both of our reconstructions. To strengthen our arguments we have now also investigated using initial object guesses that have randomly distributed imaginary components. This in effect produces random variation in both the amplitude and magnitude of the initial objects. Bowing to the experience of the PtychoShelves developers, we used the 'random' model embedded within their codebase. This has a relatively small (10^{-6} relative to the real part) but non-negligible random imaginary component. This does produce a change in our FRC characteristics, in general producing an improved characteristic, as measured by the area under the curve. However, the point of maximal crossing of the FRC with the $\frac{1}{2}$ bit threshold remains the same, so our information resolution limit by this measure remains the same and our claim is unaltered.

To explore in more detail what might be the best amount of random phase to initially introduce, or perhaps indeed to question whether an initial flat phase object is maybe a fair first approximation and reasonable piece of prior information to put into our model, would be a subject of another paper. It could perhaps be argued that a random initial phase guess is closer to the realities of our experimental sample (which have significant amounts of amorphous carbon and contamination), than the flat object we had initially used which lead to our overall improvements in the FRC, but again we'd need more investigations to prove that. Furthermore, we found that introducing an aperture constraint for the Au / aC sample (see point 2), provided an improvement in the probe model plausibility which must introduce an improvement in the reconstruction fidelity. The additional FRC characteristic is now presented in a revised version of Figure 2 show these characteristics.

The changes we have made in manuscript related to the above are:

Methods, Page 24:

... In addition, for the Au/aC reconstruction, we investigated using an initial object guess that had a flat (constant) phase and amplitude, and an initial object, O_i , with constant real component a small random imaginary component, where $\max(\text{Im}(O_i))/\text{Re}(O_i) = 10^{-6}$. This random initial object also produced a small improvement to the FRC characteristic. For the Au/MoS₂ reconstruction we used a

constant amplitude and phase initial object. Though a random object here may have improved this reconstruction too, we did not investigate it further as here we had insufficient data to perform an FRC analysis.

Main text, Page 7/8:

Here, we see (Fig. 2) that the FRC characteristic corresponding to the Au {135} plane spacing diffraction peak ring at 1.45 \AA^{-1} ($= 0.689 \text{ \AA}$) clearly correlates above the threshold, with and without a constraint on the probe FT and with constant and random initial object guesses (see Methods).

Figure 2 is now recalculated and includes an additional curve. The caption amendments are below:

Figure 2 – **Fourier ring correlation and reconstruction radial profiles from gold on amorphous carbon.** (Blue ~~ack~~ line) Average radially summed profile of the FT of the gold on amorphous carbon reconstruction given in Fig. 1(c); (dashed black ~~ue~~ line) model radial diffraction intensities expected for randomly oriented gold particles, labelled with dominant diffraction plane indices for each peak; (red line) Fourier ring correlation (FRC) measures obtained from comparison of two independently determined reconstructions, each using half of the data used to form the reconstruction in Fig. 1; (dashed orange line) FRC from a reconstruction using an aperture on the probe FT and an initial random phase object; and (green line) the half-bit resolution threshold. The maximal crossing of the FRC characteristic with the threshold gives the resolution as 0.67 \AA .

2) Extended Fig. 3 shows lattice features in the probe mode #2, while Ext. Fig. 4 which used additional probe constraints does not show these lattice features. The mode is also quite strong with 19.2%. Did a probe constraint on the Au/aC sample not improve the reconstruction?

>> Also, thank you again for your keen eye here. Our initial investigation of this probe mode looked at its FT (now presented over a larger angular range than was previously the case in Extended Figure 3) and on first inspection we did not immediately recognize any features that looked lattice-like, at least in the same way as is the case for the MoS₂ probe which obviously has lattice features. However, looking in more detail now, taking on-board the reviewer's comments, we do see apparently randomly oriented collections of 'fringes' or features that have a spacing approximately equal to the spacing of the Au 111 and 113/222 planes. This is supported by weak ring-like bands in the FT of the second probe mode (when no aperture constraint is applied). Noting this, yes, we would expect a constraint of the probe FT to improve the reconstruction.

Thus, we have performed additional reconstructions to investigate this effect introducing an aperture constraint. The probe modes for this new reconstruction are now in a modified Extended Data Figure 3, where the caption has changed thus:

Extended Data Figure 3 – **Probe models produced from the Au/aC reconstruction**. Here 3 probe modes were used and the fraction of the intensity in each mode is given at the top of each column in the table as the ratio $I_{\text{mode}}/I_{\text{total}}$. for reconstructions with (a) no aperture constraint and (b) with an aperture constraint. Each column presents a mode, and the upper rows of each table sub-part (a, b) presents :(rows 1 and 3, respectively) the amplitude and phase of the complex wavefunction representing the probe at the sample; and (rows 2 and 4, respectively) the lower row shows the amplitude and phase of the probe at the aperture plane, found by taking an FT of the complex probe, presented in rows 1 and 3.

Note that although we have removed phase information upon the probe from this figure, the phase information is still presented within the manuscript. The phase information appears (as before) in EDF 5. Previously there was some repetition and representation of the phase information between EDF 3 and EDF 5.

Thus, in our discussion of the reconstruction Au on aC we now say, on Page 7:

An example ptychographic reconstruction created from 20 keV electron diffraction data collected from gold particles supported on an amorphous carbon (Au/aC) thin film is presented in Fig. 1. To produce a physically valid sample object model, we used a multi-slice ptychographic reconstruction algorithm operating upon distortion-corrected diffraction data accompanied by a circular aperture constraint on the FT of to produce a physical probe model (see Methods)

Figure 1 has now been remade with this improved reconstruction. We also took this opportunity to make re-prepare this figure with a perceptually more uniform color-scale. Specifically, the ‘RomaO’ colormap from <https://doi.org/10.1038/s41467-020-19160-7> is used, which in comparison to our previous hsv mapping reduces or removes the appearances of artificial perceived boundaries. We hope this helps the communication of our data. The amendment to the caption of Figure 1 is below:

... enlarged region of the reconstructed exit-wave, showing the color scale used to represent the phase and amplitude. The color scale for a and b is in the lower right corner of b, where the color wheel’s azimuthal angle, ϕ , represents the phase (spanning $0 - 2\pi$) and the radial coordinate represents the amplitude of the wave, spanning an amplitude of 0.43 – 2.35. **c**, Amplitude of the Fourier transform (FT) of the intensity of the reconstructed exit wave. In **c** the intensity is gamma adjusted by 0.9, the upper and lower 1% of pixels are saturated, and the color scale which linearly represents the amplitude^{0.9} is inset at the lower right. Scalebars: **a** – 10 nm, **b** – 5 nm, **c** – 1 Å⁻¹. The hue–value color scale representing phase (hue) and amplitude (value) of the complex data inset within (b) applies also to (a).

Regarding the Au on MoS₂ reconstruction, described on page 10, our description is also thus modified with a significant section now deleted:

To form a convergent and physical ptychographic reconstruction (Fig. 3a, b), as with the Au/aC reconstruction we found it necessary to force pixels outside a certain radius in the FT of the electron probe model towards zero to gain a stable, convergent and physical probe and object reconstruction (see Methods). ~~This constraint on the electron probe model, which was not required for the Au/aC reconstruction, corresponds to the beam being defined physically by a circular aperture. The reduced data diversity in the MoS₂-based diffraction data accounts for the need to apply this additional constraint; without it the reconstruction algorithm tends to assign a non-physical object-related periodic structure to the probe.~~ Regarding the probe reconstruction, we note that ptychography in its simplest and first developed forms assumes a single coherent illumination mode propagates elastically through the sample¹⁴. In practice, real illumination systems have limited coherence and some inelastic scattering occurs in the sample. Thankfully,....

Our method description has now also been modified to include the new calculation, on page 24:

To gain successful reconstructions in the MoS₂ sample, avoiding crystalline sample related artefacts in the reconstructed electron probe, we found it necessary to constrain the probe's FT to be within an angular range less than the innermost extent of the first order diffraction discs. Given our probe-illumination angle, this constraint was reasonable and physical, simply confining the beam to have its known circular aperture constrained form. A similar constraint on the FT of the probe was applied during the Au/aC dataset reconstruction. This was found to improve the physicality of the reconstructed probe, by eliminating the presence of weak randomly oriented lattice like fringes in the second probe mode that appeared in the probe reconstruction without this constraint. Though the presence of these fringes does not alter the FRC resolution measure limit, it reduced the total area under the FRC characteristic in Fig. 2 and reduces the perceived clarity of fringes in the reconstruction (see Extended Data Fig. 11).

Here we see that this constraint on the primary mode for Au/aC reconstruction forces all the intensity within the aperture constraint. The secondary mode now appears much more diffuse and has a FT which we speculate might be related to the radial distribution function of amorphous carbon. However, to confirm that would require a further study beyond the original scope of this manuscript.

The above now references a new extended data figure (EDF 11) that presents aperture constrained and non-aperture constrained reconstructions. The differences are subtle, but as

would be expected greater clarity is seen in the fringes of the aperture constrained reconstruction here.

3) The new Extended Data Fig. 8: Yes, the figures h-j show that this spatial frequency can be resolved, but they show also exactly my concern: the atomic peaks in the 5mrad reconstruction are shifted with respect to the Ground Truth. The peaks in the 20mrad reconstruction match the ground truth.

The authors write in their rebuttal: ". Though this does not fully address the question posed, it gives some indication that when there are many spatial features present in the object (and so the object has a rich spatial diversity), accurate localization of isolated features appears possible."

Given Extended Fig. 8 I would object and say it is highly doubtful that isolated features can be localized accurately.

The text does not mention the fluence used to produce Extended Data Fig. 8 - was the fluence for the 20mrad and the 5mrad case the same and what was the fluence? The 20mrad data accurately localizes the atomic features while the 5mrad data does not.

Maybe one could show that the position of the isolated atoms is very weakly encoded in the 5mrad diffraction pattern and stronger encoded in the 20mrad diffraction pattern by comparing diffraction patterns from probe positions that are a few Angstroms apart. The 20mrad patterns should change much more strongly for small position changes.

>> The intention of our simulations was to remind readers that our use (defocused probe) ptychography here has many similarities to HR-TEM (as opposed to HR-STEM). In HR-TEM, typically the illumination convergence half angle is of order 1 mrad (or less), and sub-Å resolution is readily achievable in bright-field imaging, as Spence provided examples and the theory surrounding this in his book (High Resolution Electron Microscopy, 4th edition (or earlier ones), Section 6.2, "Single Atoms in Bright Field" for example). However, accurate localization requires consideration of contrast transfer function.

In our simulations and reconstructions there is yet another complication: Each simulated diffraction dataset was produced with a different (but known) set of random offsets from a regular monotonic grid of defocused probe positions. The reconstruction process then assigns the left-most (random) probe position to zero origin of the pixels in the reconstruction. Thus, when we compare the reconstructions with the source ground truth image there is a possibility of small offsets arising from this, despite our realignment process. With hindsight we should have set the random offset to be exactly the same in each case, or set some the extremal scan position to be *exactly* the same. The point of our reconstructions in EDF 8 was highlight the ability to resolve, and to highlight the plausibility of accurate localization.

Our results show the peak value occurring in the same pixel, but - yes – likely peak fitting would indicate there is a 5 – 10 pm shift in location for 5 mrad case versus the 20 mrad case. At this stage owing to the above-mentioned random probe-positioning and reconstruction alignment issue, we are not sure if we can trust our digital image alignments to better than +/- 10 pm accuracy. To make detailed consideration of this effect would have to be the subject of another piece of work, as we would have to trace through other numerical possibilities that may lead to a few picometres of error in positional error. Ideally some analytical treatment would be best,

keeping in mind that any real object is not a simple delta function, and hence has a non-trivial Fourier transform.

Owing to the above issues, we're uncomfortable ascribing the apparent misalignments or position error, solely to the beam half angle. For now though we feel the best we can do is to state:

Simulations presented in Extended Data Figure 8 (h-k), further support this finding by showing that two isolated peak-like features, separated by around 67 pm can be resolved in a (infinite dose) reconstruction, ~~even~~ when using a beam convergence angle of ≤ 10 mrad with a 20 keV electron beam, such as was used in our experiments. Future higher accuracy numerical and analytical work may reveal the variation and origins of any residual positional inaccuracy which is beyond, or in a similar range as, the approximate ± 10 pm registration accuracy between our simulated ptychographic reconstructions.

4) VERY IMPORTANT: The code for the main result, i.e. the experimental reconstruction, is currently not included in the paper and just "Available on request". This is completely unacceptable for an algorithm-focused paper. The code to reproduce the experimental reconstructions must be shared openly.

The full code we have used to produce our results (included the new results and analysis requested above) is now provided at <https://github.com/ArthurBlackburn/PtychoRunner>. This is accompanied by scripts to recreate all the major results of this work, i.e. reconstruction, FRC measures, PCTF, SNR, and other simulations. These codes also provide more functionality to PtychoShelves package and constitute considerable body of our current and ongoing work.

Considerable time was required to prepare the codes into a form that could be publicly shared, without disclosing further advances in our codes that might be the subject of future publications. Owing to this and other commitments, we apologize for the delay in getting this response returned.

We have also shared all the source data and reconstructions presented in this work, along with data related additional intermediate reconstruction steps. The dataset will be freely and publicly available after 16 August 2025, the current embargo period we have requested in order to complete this review. The dataset title and public doi is as below.

Arthur M. Blackburn, (2025), Electron Diffraction Data and Ptychographic Reconstructions Using a 20 keV Electron Beam on Supported Gold Nanoparticles, Federated Research Data Repository, doi: <https://doi.org/10.20383/103.01357>.

Prior to the embargo end date, reviewers are welcome to review the data using the private links which should have been forwarded to you directly from the journal editor.

Note, that the total volume of data is > 100 GB, even when using data compression. The changes made are as below (Page 14 of the revised manuscript):

Data Availability

The raw and distortion corrected experimental diffraction data, synthetic diffraction data, and all the reconstruction outputs presented in this study, along with some additional outputs from intermediate

reconstruction steps is freely available from the Canadian Federated Research Data Repository at <https://doi.org/10.20383/103.01357>. Data sets generated during the current study are available from the corresponding author on reasonable request.

Code Availability

The core PtychoShelves package used for the reconstructions, was developed by the Science IT and the coherent X-ray scattering (CXS) groups, Paul Scherrer Institute, Switzerland and is available at <https://www.psi.ch/en/sls/csaxs/software>. The multi-slice adaptation of PtychoShelves for electron microscopy used here called PtychoShelves_EM, was developed by members of Cornell University, *et al.*, and is available at <https://doi.org/10.5281/zenodo.4659690>. Code developed at the University of Victoria, to wrap PtychoShelves EM and, include ~~additional constraints~~the aperture constraint, and provide the analysis performed here is freely available at <https://github.com/ArthurBlackburn/PtychoRunner>. ~~from the corresponding author on reasonable request.~~ Prismatic software, used for producing simulated data, was developed by members of Lawrence Berkeley National Laboratory, *et al.* (see SI section 4), and is available at <https://prism-em.com/>. Our wrapper code for operating Prismatic for ptychographic data simulation is available at <https://gitlab.com/mrfitzpa/prismatique>.

5) Why did the authors opt to show only exit waves of the final model instead of the summed phase images of the final model, as now customary in multi-slice ptychography reconstruction, e.g. the original paper by Chen 2021 and the new ref 20-22? Some amplitude images are shown, but no phase images. Does phase unwrapping cause problems?

We have now tried to explain this better by offering some more possible explanations, though again this could still do with further exploration in perhaps another publication. However, directly answering the question above, we did not see any particular issues with unwrapping of individual slices before preparing the phase sum. It is hard to tell whether phase unwrapping algorithms have performed ‘perfectly’ on experimental data, but by observations there did not appear to be majorly incorrect unwrapping that would mislead our interpretations. The additional material in the **Supplementary Methods** give more of our thoughts on this subject and are below (see *Section 3: Propagated Exit Wave and Summed Phase Comparisons*):

We also propose that our use of a propagated wave as opposed to summed phase to gives the best representation of the sample and the expected trend in diffraction plane intensities, due to us performing our multi-slice reconstruction with a slice spacing (Main Text, Methods, Eqn. 3) that is towards the allowable upper limit for a valid reconstruction⁵¹. In this case the individual slices of the reconstruction

have a less directly interpretable meaning than is the case when the slice separation is significantly below this limit. When the slice separation is well below this limit, as has been used in most current high-resolution ptychographic studies of crystalline samples, the summed phase of the slices appears to match well with that expected of the system. When operating with slice thickness towards the upper limit, one may consider the individual slices as a means to produce accurately matched diffraction data, but the depth resolved information they convey is not so clear. A little more precisely, one could perhaps imagine that the coarsely separated slices of our model contain a sum of Fresnel propagated waves scattered from the atomic structure potentials that would be more clearly evident in imaginary finer separated slices that represent the sample placed between our coarsely separated slices.

This may help explain why propagating a plane wave through our slices gives an exit wave with expected trend in diffraction plane intensities, whereas the summed phase does not match as well with expectations. There does not appear to have been any problem with unwrapping the phases of the individual slices before we performed the phase summing, to compute our phase summed images and their FTs, which might have been a source of discrepancy between our models and experimental reconstructions. To aid further comparison we have also presented images of the summed phase of regions of our sample in Extended Data Figure 11. As would be expected from the FTs presented in Extended Data Fig. 6, the visibility and clarity of high-resolution information is diminished in the summed phase presentation.

We have now referenced this discussion in the main text, page 7, as below:

CTEM. ~~Also,~~ these exit-wave images produce a FT that has the expected sequence of diffraction ring intensities for randomly oriented particles, in contrast to summing the projected phase from all slices (as discussed further in Supplemental Methods, section 3, with comparisons presented in ~~and~~ Extended Data Figs. 6 and 11).

Please see the new Extended Data Figure 11.

--

Reviewer #1 (Remarks on code availability):

Currently only the code for the simulations is shared.

The code for producing the main experimental result must be shared openly and not only "on request".

» Please see our response to point 4 above.

Reviewer #2 (Remarks to the Author):

The authors have addressed most of the issues raised by the referees. The paper is now rather long by the standards of printed articles, but a brief look at Nature Comms suggests that this is quite normal. Some questions have not been answered fully, but only because to do so would require a very substantial additions to paper. In such instances, the authors have added to the text to highlight these complications. By leaving questions open, the paper invites others to think about, and work on, these matters, which is a good thing. (As an example - the effect on the fidelity of the reconstruction in the presence of a weakly scattering amorphous support overlaying a dominant periodic feature.) But it would over-burden the present work to cover all these nuances – it would also require a lot more work beyond the main scope of the paper. I think it is thorough enough as it stands.

» Thank you for your comments. Certainly the questions this paper raises will lead to some interesting future work for us, particularly in the topic we have highlighted in your comment above.

Reviewer #3 (Remarks to the Author):

The authors have clearly made significant efforts to address the comments raised in my review of their initial submission. Having read their responses to all of the referees comments I am now happy that the manuscript can be published. My only remaining minor point is that I feel that basing the use of an uncorrected SEM for Ptychography on cost grounds only tells part of the story - the availability of a large volume around the sample which could be used to introduce external stimuli more conveniently and flexibly than in a TEM is surely equally important.

» Thank you also. Yes, that's also an interesting and important area that using an SEM opens up. We had unfortunately omitted mention of that, but have now noted that in the revised text.

See the revised introduction, page 4:

Such devices still require more research along with a scaling of production and associated observation techniques to meet required levels for manufacturing readiness³⁷. Furthermore, SEM instruments which typically operate at ≤ 30 keV have much greater chamber space around the sample in comparison to TEM, which facilities applying external stimuli and performing in-situ experiments. Thusesese , many materials, -and technologies an in-situ atomic scale investigations would thus-also benefit from sub-Å imaging at ≤ 30 keV.

Also see the addition to the discussion, page 14:

Given the sample thickness constraints of lower beam energies, we anticipate our advance to most immediately benefit research and production of 2D material-based devices³⁷-, and facilitate in-situ atomic scale investigations within the larger sample chamber of typical SEM instruments.

Thank you again for pointing this out.

Other small changes

In re-reading our manuscript a few other small corrections have been made:

Methods: Page 23

The reconstructions for the Au/aC (Au/MoS₂) samples used 50 iterations of ePIE followed by 300 iterations of LSQ-ML, and 600 (1~~7~~600) iterations of MS-LSQ-ML.

The numbers in Extended Data Table 1 remain as they were and are correct. In our previous methods notes, we made a 'manual addition' error which, where we were off by 1 digit. We have now corrected this.

Methods: Page 24

Here the fitting was performed by minimizing the Euclidean distance between the experimental and model radial profiles, using the simulated annealing algorithm, though we also found that genetic algorithms and pattern search methods yielded very similar results.

Our code on Github points this out. It seems worthwhile to mention it here too.

To be more in-line with the Nature Communications heading style (which appears to have no sub-levels), we took our existing headings and combined them. Though this might be an editorial choice, we thought it prudent to suggest a suitable change here:

Page 4:

Results

Results: Diffraction Data Distortion Correction

Page 7

Results: Gold on Amorphous Carbon Reconstruction

Page 10

Results: Gold on MoS₂ Reconstruction

We also corrected 2 instances where we had referred to 'Extended Methods', whereas in fact our additional material is called 'Supplemental Methods'.